


# Challenging the timely prediction of landside early warning systems with multispectral remote sensing: a novel conceptual approach tested in the Sattelkar, Austria

Doris Hermle[1], Markus Keuschnig[2], Ingo Hartmeyer[2], Robert Delleske[2] and Michael Krautblatter[1]

[1]Technical University of Munich, Chair of Landslide Research, Munich, Germany
[2]GEORESEARCH Forschungsgesellschaft mbH, Puch, Austria

*Correspondence to*: Doris Hermle (doris.hermle@tum.de)

**Abstract**

While optical remote sensing has demonstrated its capabilities for landslide detection and monitoring, spatial and temporal

demands for landslide early warning systems (LEWS) were not met until recently. We introduce a novel conceptual approach for comprehensive lead time assessment and optimisation for LEWS. We analysed "time to warning" as a sequence; (i) time to collect, (ii) to process and (iii) to evaluate relevant optical data. The difference between "time to warning" and "forecasting window" (i.e. time from hazard becoming predictable until event) is the lead time for reactive measures. We tested digital image correlation (DIC) of best–suited spatiotemporal techniques, i.e. 3 m resolution PlanetScope daily imagery, and 0.16 m

resolution UAS derived orthophotos to reveal fast ground displacement and acceleration of a deep–seated, complex alpine mass movement leading to massive debris flow events. The time to warning for UAS and PlanetScope totals 31h/21h and is comprised of (i) time to collect 12/14h, (ii) process 17/5h and (iii) evaluate 2/2h, which is well below the forecasting window for recent benchmarks and facilitates lead time for reactive measures. We show optical remote sensing data can support LEWS with a sufficiently fast processing time, demonstrating the feasibility of optical sensors for LEWS.

**1 Introduction**

Landslides are a major natural hazard leading to human casualties and socio–economic impacts, mainly by causing infrastructure damage (Dikau et al., 1996; Hilker et al., 2009). They are often triggered by earthquakes, intense short–period or prolonged precipitation, and human activities (Hungr et al., 2014; Froude and Petley, 2018). Gariano and Guzzetti (2016) report in a review study that 80 % of the examined papers show causal relationships between landslides and climate change.

The ongoing warming of the climate (IPCC, 2014) is likely to decrease slope stability and increase landslide activity (Huggel et al., 2012; Seneviratne et al., 2012). This indicates a vital need to improve the ability to detect, monitor and issue early warnings of landslides and thus to reduce and mitigate landslide risk. Early warning, as defined by the UN International Strategy for Disaster Reduction (UNISDR), refers to a set of capacities for the timely and effective provision of warning information through institutions, such that individuals, communities and organisations exposed to a hazard are able to take

action with sufficient time to reduce or avoid risk and prepare an effective response (UNISDR, 2009). This definition of an early warning system (EWS) contains a time component but includes no exact time scale reference. 'Early' suggests that events are detected before harm or damage occurs and thus stands in contrast to events which are only detected once they have begun (e.g. snow avalanches). Thus, it is necessary to know sensor capabilities and limitations for pre–event mass movement observations (Desrues et al., 2019). The success of a warning requires that information is provided with enough lead time for

decisions on reactions and counter measures (Grasso, 2014). According to UNISDR (2006), an effective early warning system consists of four elements: (1) risk knowledge, the systematic data collection and risk assessment; (2) the monitoring and warning service; (3) the dissemination and communication of risk as well as early warnings; and (4) the response capabilities on local and national levels. Incompleteness or failure of one element can lead to a breakdown of the entire system (ibid.).



This study presents a new concept to systematically evaluate remote sensing techniques to optimise lead time for landslide
early warnings. We do not start from the perspective of available data; instead, we define necessary time constraints to
successfully employ remote–sensing data for providing early warnings. This approach reduces the number of suitable remote
sensing products to a small number with high temporal and spatial resolution. With these constraints, we investigated the
application of data from satellites and unmanned aerial systems (UAS) to allow the assessment of the data, after a spaceborne
area–wide but low–resolution acquisition, into a downscaled detailed image recording. In so doing, we analysed the capability
of these different passive remote sensing systems focusing on spatiotemporal capabilities for ground motion detection and
landslide evolution to provide early warnings.

Until recently, the spatial and temporal resolution requirements for accurate early warning purposes have not been met by
optical satellite imagery (Scaioni et al., 2014). This is essential since temporal resolution determines whether landslide
monitoring allows defining displacement rates and enables approximating acceleration thresholds, which are lacking if
information is based solely on post–event studies (Reid et al., 2008; Calvello, 2017). Landslide monitoring therefore not only
deepens the understanding of landslide processes but also has the potential to significantly advance landslide early warning
systems (LEWS) (Chae et al., 2017; Crosta et al., 2017). Previously, high spatial resolution satellite data was obtained at the
expense of a reduction in the revisit rates (Aubrecht et al., 2017). Consequently, the return period between two images
increased, limiting ground displacement assessment and the range of observable motion rates. The number of useful images
was further reduced due to natural factors such as snow cover, cloud cover and cloud shadows. High–resolution remote sensing
data was long restricted due to high costs and data volume (Goodchild, 2011; Westoby et al., 2012). Commercial very high
resolution (VHR) optical satellites exist, but tasked acquisitions make them inflexible and very cost intensive, thus limiting
research (Butler, 2014; Lucieer et al., 2014). There is a vast spectrum of available remote sensing data with high spatiotemporal
resolution (Table 1). Complementary use of different remote sensing sources can significantly improve landslide assessment
as demonstrated by Stumpf et al. (2018) and Bontemps et al. (2018), who draw on archive data and utilise different sensor
combinations to analyse the evolution of ground motion.

**Table 1** Overview of different optical multispectral remote sensors with their corresponding resolution [m] and revisit rate [days]. The
sensors are categorised into commercial and free data policy. [1]free quota via Planet Labs Education and Research Program, [2]PlanetScope
Ortho Scene Product, Level 3B/Ortho Tile Product, Level 3A (Planet Labs, 2020b), [3]reached end of life, 3/2020, archive data usable, [4]5 m
Ortho Tile Level 3A (Planet Labs, 2020a), [5]0.5 m colour pansharpened, [6]self–acquired. Source: (ESA, 2020).

| Sensor | Temporal resolution [d] | Spatial resolution [m] | Free/ Commercial |
|---|---|---|---|
| UAS | flexible | 0.08 | F[6] |
| WorldView 2 | 1.1 | 1.84 | C |
| WorldView 3 | <1 | 1.24 | C |
| WorldView 4 | <1 | 1.24 | C |
| GeoEye 2 | 5 | 1.24 | C |
| SkySat | 1 | 1.5 | C |
| GeoEye–1 | 3 | 1.64 | C |
| Pléiades 1A/B | 1 | 2.0 (0.5)[5] | C |
| PlanetScope | 1 | 3.0/3.125[2] | C/F[1] |
| RapidEye[3] | 5.5 | 5[4] | F |
| Sentinel–2 A/B | 5 | 10 | F |
| Landsat 8 | 16 | 30 | F |

The latest developments in earth observation programs include both the new Copernicus' Sentinel fleet operated by the ESA,
and a new generation of micro cube satellites, sent into orbit in large numbers by PlanetLabs Inc. These PlanetScope micro
cube satellites, known as 'Doves', and Sentinel–2 a/b offer very high revisit rates of 1–5 days and high spatial resolutions from
3–10 m, respectively (Table 1), for multispectral imagery (Drusch et al., 2012; Butler, 2014; Breger, 2017). This opens up
unprecedented possibilities based on these high spatiotemporal resolutions to study a wide range of landslide velocities and
natural hazards through remote sensing. Future data access is fostered by PlanetLabs and by Copernicus (via its open data



policy) providing affordable or free data for research. This leads to unprecedented possibilities for studying natural hazards

through remote sensing. Examples of such multi–temporal studies of landslide activities based on this access to high spatiotemporal data are Lacroix et al. (2018), using Sentinel–2 scenes to detect motions of the 'Harmalière' landslide in France, and Mazzanti et al. (2020), who applied a large stack of PlanetScope images for the active Rattlesnake landslide, USA.

As forecasted landslides tend to accelerate beyond the deformation rate observable with radar systems before failure, we concentrate on optical image analysis. One advantage of optical imagery is its temporally dense data (Table 1) compared to

open data radar systems with sensor visits more than every six days (Sentinel–1, ESA). Optical data allows direct visual impression from the multispectral representation of the acquisition target and the option to employ this data for further complementary and expert analyses. While active radar systems overcome constraints posed by clouds and do not require daylight, data voids can be significant due to layover or shadowing effects in steep mountainous areas (Moretto et al., 2016; Plank et al., 2015). Moreover, north/south facing slopes are less suitable, thus limit the range of investigation (Darvishi et al.,

85 2018).

In general, sensor choice depends on the landslide motion rate with radar at the lower and optical instruments at the upper motion range (Crosetto et al., 2016; Moretto et al., 2017; Lacroix et al., 2019). A flexible, cost–effective alternative to spaceborne optical data are airborne optical images taken by UASs (unmanned aerial system). Freely selectable flight routes and acquisition dates prevent shadows from clouds and topographic obstacles, and allow avoiding unfavourable weather

conditions and summer time snow cover, all of which frequently impair satellite images (Giordan et al., 2018; Lucieer et al., 2014). UAS–based surveys provide accurate very high resolution (few cm) orthoimages and digital elevation models (DEM) of relatively small areas, suitable for detailed, repeated analyses and geomorphological applications (Westoby et al., 2012; Turner et al., 2015).

In recent years, data provision for users has increased and today data hubs provide easy accessibility to rapid, pre–processed

imagery. Knowledge of the most useful remote sensing data options is vital for complex, time–critical analyses such as ground motion monitoring and landslide early warning. Nonetheless, technological advances can be misleading as they promise high spatiotemporal data availability, which frequently does not reflect reality (Sudmanns et al., 2019). One key problem is the realistic net temporal data resolution which is often significantly reduced due to technical issues, such as image errors and non–existent data (i.e. data availability, completeness, reliability). Other problems include data quality and accuracy in terms

of geometric, radiometric and spectral factors (Batini et al., 2017; Barsi et al., 2018). Timely information extraction and interpretation are critical for landslide early warning yet few studies have so far explicitly focused on time criticality and the influence of the net temporal resolution of remote sensing data.

In this investigation we propose both a conceptual approach to evaluating lead time as a time difference between the "time to predict" and the forecasting time and assess the suitability of UAS sensors (0.16 m) and PlanetScope (3 m) imagery (the latter

with temporal proximity to the UAS acquisition) for LEWS. For this we have chosen the 'Sattelkar', a steep, high–alpine cirque located in the Hohe Tauern Range, Austria (Anker et al., 2016). We estimate times for the three steps (i) collecting images, (ii) pre–processing and motion derivation by digital image correlation (DIC) and (iii) evaluating and visualizing. The results from the Sattelkar site – and from historic landslide events – will be discussed in terms of usability and processing duration for critical data source selection which directly influences the forecasting window. Accordingly, we try to answer the following

research questions:

1. How can we evaluate lead time as a time difference between the "time to predict" and the forecasting time for high spatiotemporal resolution sensors?

2. How can we quantify "time to warning" as a sequence of (i) time to collect, (ii) to process and (iii) to evaluate relevant optical data?

3. How can we practically derive profound "time to warning" estimates as a sequence of (i), (ii) and (iii) from UAS and PlanetScope high spatiotemporal resolution sensors?



4. Are estimated "times to warning" significantly shorter than the forecasting time for recent well–documented examples and able to generate robust estimations of lead time available to enable reactive measures and evacuation?

## 2 Lead time – a conceptual approach

### 2.1. The conceptual approach

Natural processes and natural developments constantly take place independently, thus dictate the technical approaches and methodologies researchers must apply within a certain time period. For that reason, we hypothesise the forecasting window $t_{external}$ is externally controlled, consequently the applicability of LEWS methods ($t_{internal}$) is restricted because they must be shorter than $t_{external}$. This approach is the framework of our time concept (Fig. 1).

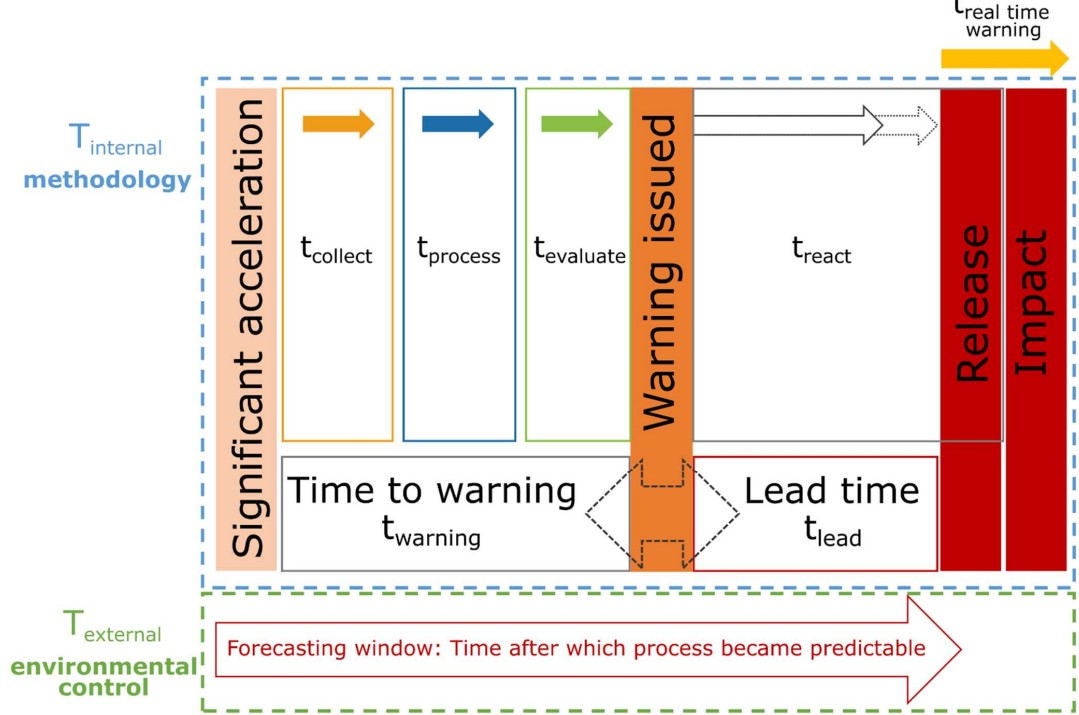


**Figure 1** The novel conceptual approach for lead time, time to warning and the forecasting window for optical image analysis.

The forecasting window is started ($t_{external}$, green dashed outline) following significant acceleration exceeding a set displacement threshold, leading to a continuous process. Simultaneously with the forecasting window, time to warning ($t_{warning}$) starts (grey outline). Time to warning is divided into a three–phase–process to allow time estimations for a comparative

assessment of different types of remote sensing data. This process consists of the phases (1) time to collect, (2) time to process and (3) time to evaluate, each with their individual durations. Confidence in the forecasted event increases with time as process acceleration becomes more certain. Once a warning is released (orange box), the lead time begins ($t_{lead}$) and is terminated by the following release and subsequent impact (red box). The lead time is the difference between the forecasting window and the time to warning. During the lead time, reaction time ($t_{react}$) starts when appropriate counter measures are taken to prepare

for and reduce risks ahead of the impending event, and ends with the final impact.

The time to warning period ($t_{warning}$) is defined by the time necessary to systematically collect data, analyse the available information and to evaluate it. Hence, the greater the lead time, the more extensive countermeasures can be implemented prior



to the event. An imperative for an effective EWS, the required time to take appropriate mitigation and response measures has to be within the lead time interval ($t_{lead}$) (Pecoraro et al., 2019) with $t_{lead} \geq t_{react}$ .

**2.2. General applicability to optical data**

The time to warning consists of a three–phase–process (see Sect. 2.1. and Fig. 1) to allow rough time estimations for a comparative assessment of different types of remote sensing data. Nevertheless, to realise this temporal concept an established, operating system is required, which includes reference data (DEM, previous results), experience from past field work and ready UAS flight plans with preparation for a UAS flight campaign, satellite data access, experience in the single software processing steps including final classification and visualisation templates and, if utilised for UAS, installed and measured ground control points.

The first phase includes the collection of data starting from the acquisition by the sensor, the data transfer, image pre–processing and provision to the end user. The user selects images online from the data hub, downloads and organises them. For a UAS campaign, the user must obtain flight permits, check flight paths and conduct the UAS flight. The second phase encompasses time to process for the complete data handling from the downloaded data to final analysis–ready image stacks in a GIS or a corresponding software. These preparatory steps may include image selection and renaming, atmospheric correction, co–registration, resampling and translation to other spatial resolutions and geographic projection systems, adjustments such as clipping, stacking of single bands into one multispectral image or the division into single bands, calculation of hillshade from DEM among others, depending on the requirements. Following this preparation, the data is processed with the appropriate software tools to derive ground motion, calculate total displacement and derive surface changes, e.g. volume calculations or profiles. In the third and last phase, time to evaluate, the results are compared to inventory data and, if available, ground truth data, displacement results of other sensors or different spatial resolutions, different time interval variations to observe changes in sensitivity to meteorological conditions. Additionally, filters may be applied to eliminate noise. Finally, the results are analysed and evaluated. In each phase quality management is carried out for data access and pre–and post–processing. In time to collect, the images must be selected manually prior to any download from the data hub, as its filter tool options on cloud and scene coverage are of limited help. Accordingly, the areal selection may be misleading as the region of interest (RoI) might not be fully covered, though the sought–for, smaller area of interest (AoI) is covered but not returned from the request. Concerning cloud filters, first, the filter refers to the RoI as a whole in terms of percentage of cloud coverage. The AoI can still be free of clouds or else be the only area covered by clouds in the total RoI. Therefore, an image is either not returned although usable, or returned but not useable. Second, clouds can create shadows for which no filter is available. As a result, affected images have to be manually removed by the user. Images which are of low quality due to snow cover have to be discarded, too. These actions indirectly represent first quality checks in the collection phase. In the following processing phase, the images in a GIS, are checked for quality and accuracy. Depending on the data provider, some pre–processing such as radiometric, atmospheric and/or geometric corrections may have been conducted. During this phase, additional user–based steps will be checked if necessary. Finally, the results are compared to other data (e.g. DEM, dGPS), reviewed for their validity and may be supplemented by statistical evaluation.

**3 Study Site**

The Sattelkar is a high–alpine, deglaciated west–facing cirque at an altitude of between 2 130–2 730 m asl in the Obersulzbach valley, Großvenedigergruppe, Austria (Fig. 2a). Surrounded by a headwall of granitic gneiss, the cirque infill is characterised by massive volumes of glacial and periglacial debris as well as rockfall deposits (Fig. 2b, c). Near–surface temperature data indicates sporadic permafrost distribution in the upper part of the cirque. Since 2003 surface changes have taken place as evidenced by a massive degradation of the vegetation cover and the exposure and increased mobilisation of loose material. A



terrain analysis revealed that a deep–seated, retrogressive movement in the debris cover of the cirque had been initiated (Anker et al., 2016; GeoResearch, 2018). High water (over)saturation is assumed to be causing the spreading and sliding of the glacial

and periglacial debris cover on the underlying, glacially smoothed bedrock cirque floor forming a complex landslide (Hungr et al., 2014). Detailed aerial orthophoto analyses, witness reports and damage documentations indicate a steady increase in mass movement and debris flow activity over the last decade (Anker et al., 2016).

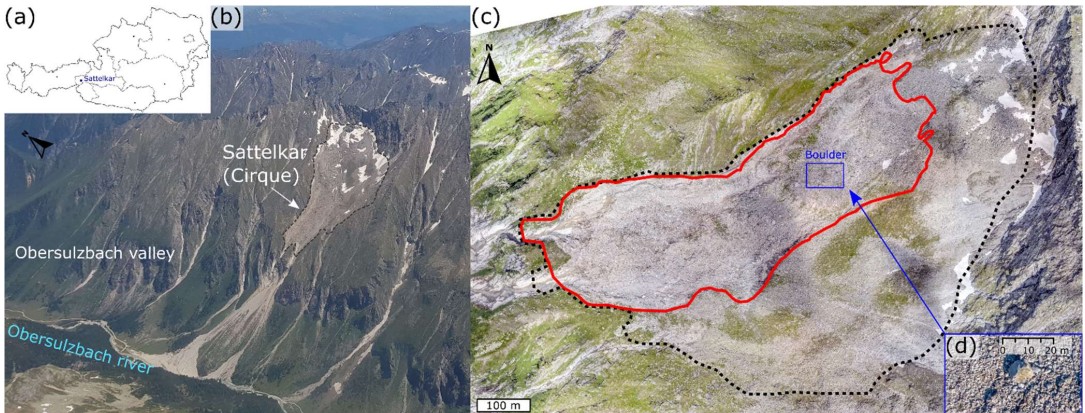

**Figure 2** (**a**) Overview map Austria (Österreichischer Bundesverlag Schulbuch GmbH & Co. KG and Freytag-Berndt & Artaria KG, Vienna). (**b**) Sattelkar, 30.6.2019 with the debris cone of the 2014 debris flow event and (**c**) UAS orthophoto (04.09.2019, 1:1.000) showing boulder sizes of 5–10 m used for manual motion tracking, (**d**) active boulder blocks from the central AoI.

In August 2014, heavy ongoing precipitation triggered massive debris flow activity of 170 000 m³ in volume, of which approximately 70 000 m³ derived from the catchment above 2 000 m. A further 100 000 m³ was mobilised in the channel within the cone. The consequence was that the Obersulzbach river was blocked leading to a general flooding situation in the

catchment, resulting in substantial destruction in the middle and lower reaches (Fig. 3).

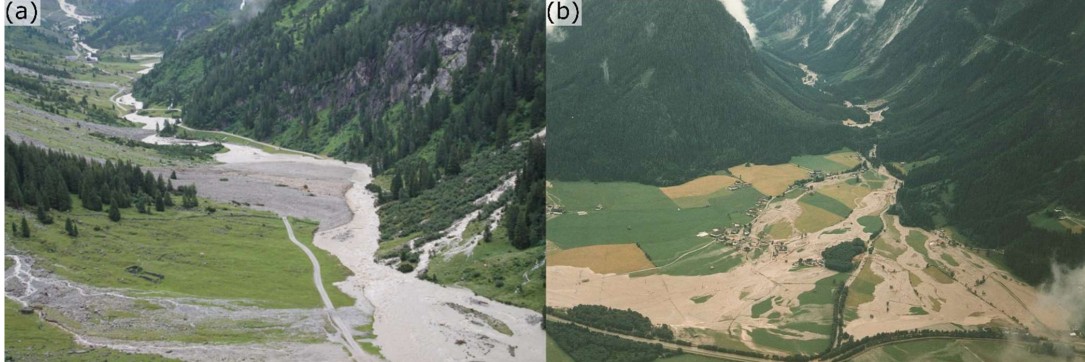

**Figure 3** Obersulzbach valley, flood event September 2014. (**a**) Flooding situation in the Obersulzbach valley with the Sattelkar landslide cone deposit (image centre). (**b**) Flood area at the valley mouth in Sulzau and Schaffau. The Salzach river is at the bottom of the image. ©Salzburger Nachrichten/Anton Kaindl.

The Sattelkar has been the focus of international research projects such as "PROJECT Sattelkar" (GeoResearch, 2018) and AlpSenseBench (TUM, Chair of Landslide Research, 2020) since 2018. In 2015 preliminary findings revealed a mass movement coverage of 130 000 m² with approximately 1 mio. m³ of debris and displacement rates of more than 10 m a⁻¹. The debris consists of boulders up to 10 m in diameter (Fig. 2c, d) allowing visual block tracking. High displacement was measured between 2012 and 2015 with up to 30 m a⁻¹.

In the Sattelkar cirque, several monitoring components are installed to provide ongoing and long–term monitoring. These components include 30 near surface temperature logger (NSTL) nine permanent ground control points (GCP) measured with





a dGPS to provide stable and optimal conditions for the derivation of orthophotos from highly accurate UAS images (GeoResearch, 2018). Field–based mapping and measurements help to delimit the active process area.

The Sattelkar is a suitable case study as it is in the early stages of the landslide development and thus fits best to this conceptual
approach. Here, processes take place on time scales appropriate for long–term observation to provide sufficient warning time. The active part of the cirque has accelerated in recent years allowing the analysis of EWS concepts based on multispectral optical remote sensing data supported by complementary block tracking.

## 4 Materials and Methods

### 4.1. Optical imagery

Optical satellite imagery is more appropriate for high deformation studies than radar applications due to the high spatial resolution as well as the short time span between acquisitions (Delacourt et al., 2007). Although the west–facing slope is favourable for the application of radar derivatives (InSAR/DInSAR), the choice to use optical imagery is based on the observed high displacement rates, which cause decorrelation when using radar technologies as they are more sensitive than optical technologies. Complex and/or large displacement gradients make the phase ambiguity difficult to solve for radar interferometry
(Kääb et al., 2017). Revisit times of current radar satellites (e.g. Sentinel–1) are longer than those of optical satellites, and if time periods between image acquisition become too long, ground motion may accumulate such that the displacement is too high to be measured. Several studies on displacements of faults and landslides have shown the potential of optical data to provide detailed displacement measurements based on image correlation techniques (DIC) (Leprince et al., 2007; Rosu et al., 2015). A further advantage of optical images for geomorphological processes in steep terrain is their viewing geometry (close
to nadir) (Lacroix et al., 2019). Here we employ DIC to compare the spatiotemporal resolution of optical imagery (UAS and PlanetScope) and to assess its suitability for early warning purposes. UAS images offer excellent spatial resolution and accuracy at the centimetre scale (Turner et al., 2015) and complement large scale satellite or airborne acquisitions (Lucieer et al., 2014). PlanetScope imagery provides the highest temporal resolution among available sensors with daily acquisitions, guaranteed data availability, and free and open access for research purposes. In this study the PlanetScope Analytic Ortho
Scene SR (surface reflectance) imagery (16–bit, geometric–, sensor– and radiometric corrections) was employed (Planet Labs, 2020b) and was supported by the Planet Labs Education and Research Program.

### 4.2. Data availability of PlanetScope

Research on the availability and usability of PlanetScope imagery was conducted on the Planet Explorer data hub for the time span from the beginning April to the end of October in 2019, as during these months snow cover should be negligible. Filter
parameters were solely set for 4–band PlanetScope Ortho Scenes and the Sattelkar AoI. In order to obtain all available images, no filters (e.g. sun azimuth, off nadir angle) were applied. We defined four categories i) meteorological constraints due to snow cover, cloud cover and cloud shadow; ii) image (coverage) errors made by the provider, iii) no data availability and iv) the remainder of usable data (Table 2). The output request was evaluated according to the defined categories and was compared to the provider's guaranteed daily image provision, which is comprised of 213 days for the time period (01.04.2019–
31.10.2019). We calculated percentages for the above categories based on days per month as well as a seven–month sum and percentage average. The availability analysis did not include an examination of the data with regard to its spatial usability: positional accuracy and/or image shifts.



**Table 2** PlanetScope 4–band data availability and usability for Sattelkar AoI for April to October 2019.

| Month | | April (%) | May (%) | June (%) | July (%) | August (%) | September (%) | October (%) | 7 month sum | 7 month avg (%) |
|---|---|---|---|---|---|---|---|---|---|---|
| usable | | 0.0 % | 0.0 | 20.0 | 22.6 | 9.7 | 13.3 | 9.7 | 23 | 10.7 |
| unusable | | | | | | | | | | |
| | cloud cover/shadow | 16.7 | 6.5 | 0.0 | 19.4 | 32.3 | 16.7 | 9.7 | 31 | 14.5 |
| | snow cover | 10.0 | 0.0 | 33.3 | 0.0 | 0.0 | 3.3 | 3.2 | 15 | 7.0 |
| | image errors | 23.3 | 25.8 | 16.7 | 12.9 | 29.0 | 20.0 | 19.4 | 45 | 21.0 |
| | no coverage/data voids | 10.0 | 12.9 | 16.7 | 32.3 | 16.1 | 20.0 | 32.3 | 43 | 20.1 |
| not available | no upload | 40.0 | 54.8 | 13.3 | 9.7 | 12.9 | 26.7 | 25.8 | 56 | 26.2 |

Unfavourable meteorological influences of cloud cover/shadow and snow cover affected up to 32.3 % and up to 33.3 %, respectively, on all 213 days; on average 14.5 % and 7 % of the days were not usable (Table 2). For 10 days in June snow influence had the greatest negative share (33.3 %), for April there were three days of snow coverage and the months September

and October each had one day of snow coverage. Cloud cover/shadow exerted a higher impact on data usability by 14.5 %. Problems on the part of PlanetLabs made much of the data unusable due to image errors; between four and nine images per month were not usable (21 %). On average for 26.2 % of the analysed time period no image data was available. In this seven–month period, 43 images (20.1 %) had data voids or did not cover the AoI.

### 4.3. Data Acquisition and Processing

In line with the concept in Fig. 1 (Sect. 1), the following processing steps are categorised and described.

(1) $t_{collect}$: UAS data acquisition was preceded by detailed flight route planning and checks of local weather and snow conditions. UAS flights were carried out with a DJI Phantom4 UAS on 13.07.2018, 24.07.2019 and 04.09.2019 (see Table 3, Fig. 4).

**Table 3** Acquisition dates of UAS and PlanetScope images, in chronological order.

| Acquisition set | UAS | PlanetScope |
|---|---|---|
| (1) | 13 July 2018 | 02 July 2018 (a), 19 July 2018 (b) |
| (2) | 24 July 2019 | 24 July 2019 |
| (3) | 04 September 2019 | 04 September 2019 |


For each acquisition, the total area was covered by four flights which were started on different elevations (Table 4). Flight planning was done with UgCS maintaining a high overlap (front: 80 %, side: 70 %) and a target ground sampling distance (GSD) of 7 cm. The area covered was approximately 3.4 km² and with a flight speed of about 8 m/s total flight time took 3.5 hours. The images were captured in RAW format. In the Planet Explorer Data Hub, PlanetScope Ortho Scenes were

selected for usability; imagery affected by snow cover, cloud cover, cloud shadow and partial AoI coverage was discarded (Table 5).

**Table 4** UAS Flight plans.

| Flight plan parts | Length of flightpath [km] | Flight time [min] | Passes | No. of images | GSD | Altitude start point [m] | Highest flight position [m] | Lowest terrain point [m] |
|---|---|---|---|---|---|---|---|---|
| Top | 6.8 | 17 | 6 | 121 | 7 | 2630 | 3120 | 2365 |
| Middle | 7.5 | 19 | 6 | 135 | 7 | 2200 | 2682 | 1820 |
| Low 1 | 7.3 | 17 | 6 | 130 | 7 | 1768 | 2115 | 1620 |
| Low 2 | 5.6 | 14 | 6 | 81 | 7 | 1768 | 2110 | 1620 |
| Total | 27.2 | 67 | 24 | 467 | 7 | | 3120 | 1620 |





**Table 5** Planet Scope Ortho Scenes.

| Acquisition Date | Identifier | Incidence Angle [deg] |
|---|---|---|
| 02.07.2018 | 20180702_093434_0f3f_3B_AnalyticMS_SR | 2.18E-01 |
| 19.07.2018 | 20180719_093512_0f3f_3B_AnalyticMS_SR | 2.36E-01 |
| 24.07.2019 | 20190724_094200_1014_3B_AnalyticMS_SR | 5.57E+00 |
| 04.09.2019 | 20190904_093632_0e20_3B_AnalyticMS_SR | 4.24E+00 |

(2) $t_{process}$: in phase two (time to process) the PlanetScope images were visualised in QGIS. Thereafter, a second selection from the downloaded images was filtered for errors of location, shift and spectral colour problems which were previously not clearly discernible in the online data hub. The final selection of images was made based on the temporal proximity to the UAS data

to guarantee the best comparability. For acquisition set (1), there are two PlanetScope images (02.07.2018 and 19.07.2018) which differed from the UAS acquisition date (13.07.2018) by 11 and 6 days, respectively. For acquisition sets (2) and (3), PlanetScope and UAS acquisition dates were identical (24.07.2019 and 04.09.2019). The acquired data sets were categorised in chronological intervals I/Ia/Ib and II (see Fig. 4). The PlanetScope images (19.07.2018, 24.07.2019 and 04.09.2019) were taken between 11:35 and 11:42 local time.


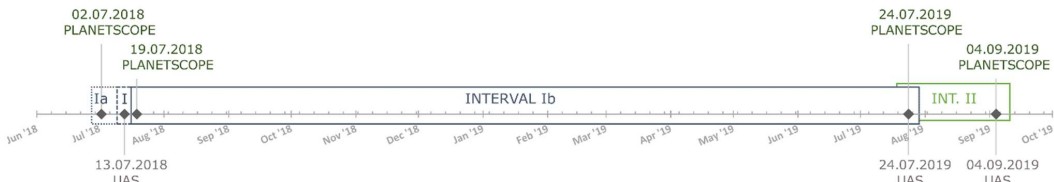

**Figure 4** Acquisition dates of UAS and PlanetScope images within the investigated time period. Calculated interval I for UAS images (13.07.2018–24.07.2019, 376 d) and interval Ib for PlanetScope images (19.07.2018–24.07.2019, 370 d), interval II for UAS and PlanetScope images (24.07.2019–04.09.2019, 42 d). Note: Ia PlanetScope interval was discarded.

The UAS images in RAW format were modified using Adobe Exposer to improve contrast, highlights, shadows and clarity. Thereafter, they were exported as JPG (compression 95 %) and processed with Pix4Dmapper to 0.08 m resolution and orthorectified based on nine permanent ground control points (GCP, 30 x 30 cm) registered with dGPS for georeferencing and further rectification of the UAS images.

Next, the data was clipped to a common area of interest (AoI) and resampled with GDAL and the cubic convolution method

to 0.16 m to enhance processing time and increased reliability of image correlation. PlanetScope Satellite images were co–registered in Matlab relative to a reference image.

We used digital image correlation (DIC) to measure the displacement for the active landslide body of the Sattelkar and to assess the suitability of the PlanetScope and UAS data. This method employs optical data and calculates the distance between an image pair, based on location changes of common pixels. The result provides displacement and ground deformation in 2 D

on a sub–pixel level. COSI–Corr (Co–registration of Optically Sensed Images and Correlation), a widely used software in landslide and earthquake studies was used for sub–pixel image correlation (Stumpf, 2013; Lacroix et al., 2015; Rosu et al., 2015; Bozzano et al., 2018). COSI–Corr is an open source software add–on developed by CALTECH (Leprince et al., 2007), for ENVI classic. Parameter settings include the initial and final window sizes in x, y; a direction step in x, y between the sliding windows; and several robustness iterations (Table 6). There are two correlators; in the frequency domain based on FFT

(Fast Fourier Transformation) and a statistical one. Applying FFT, different parameter combinations of window sizes, direction step sizes and robustness iterations were tested. We utilised recommended window sizes as suggested by Leprince et al. (2007) and Bickel et al. (2018). Step size one showed good results while keeping the original spatial resolution for the output; robustness iterations of two to four were sufficient for our purposes. Initial and final window sizes were systematically tested





(see Table 6). For computing a state–of–the–art powerstation was employed (AMD Ryzen 9 3950X 16–core processor,
3.70 GHz, 128 GB RAM).

Table 6 COSI–Corr input parameters for intervals of UAS and PlanetScope.

| Sensor Resolution | Input interval | Initial window [pix] | Final window [pix] | Robustness iteration | Step size |
|---|---|---|---|---|---|
| UAS [0.16 m] | I: 13 July 2018–24 July 2019 II: 24 July 2019–04 September 2019 | 128x128 | 32x32 | 2 | 1x1 |
| UAS [3.0 m] | I: 13 July 2018–24 July 2019 II: 24 July 2019–04 September 2019 | 32x32 | 16x16 | 2 | 1x1 |
| PlanetScope [3.0 m] | Ib: 19 July 2018–24 July 2019 II: 24 July 2019–04 September 2019 | 64x64 | 32x32 | 4 | 1x1 |

The results of the signal–to–noise ratio (SNR), east–west and north–south displacements were exported from ENVI classic as
GTiff and the total displacement was calculated with QGIS.

(3) $t_{evaluate}$: in the last phase (time to evaluate) the results of various parameter settings were compared in QGIS and ArcGIS
along with different combinations of visualisation. Displacement below a 4 m threshold was discarded from the PlanetScope
datasets due to aberrant values (noise, outliers); no other filters were employed, and we maintained the output raw. Very few
inconsistencies were present in the UAS–derived displacement results, which were accepted without modification.

Additional analyses were performed to estimate the DIC outputs of both, the UAS orthophotos and PlanetScope satellite
imagery. Visual tracking of 36 single blocks, identifiable in the UAS orthophoto series allowed deriving direction and amount
of movement; this supported the verification process of the total displacement. We employed this approach for the time
interval I. In order to assess the information value and validity of the satellite imagery, UAS orthophotos were downsampled
to 3 m (cubic convolution) for comparison purposes prior image correlation.

## 5. Results


In Sect. 5.1. we present ground motion results from DIC for the original input resolution for i) UAS, 0.16 m and ii) PlanetScope,
3 m input resolution based on parameters in Table 6. Second, for iii) DIC results of UAS downsampled to 3 m and of
PlanetScope are compared. In Sect. 5.2. DIC results for UAS, 0.16 m are analysed with regard to displacement of visual single
block tracking. Finally, in Sect. 5.3. required times for $t_{collection}$, $t_{processing}$ and $t_{evaluation}$ for each sensor are presented.

### 5.1. Displacement Rates

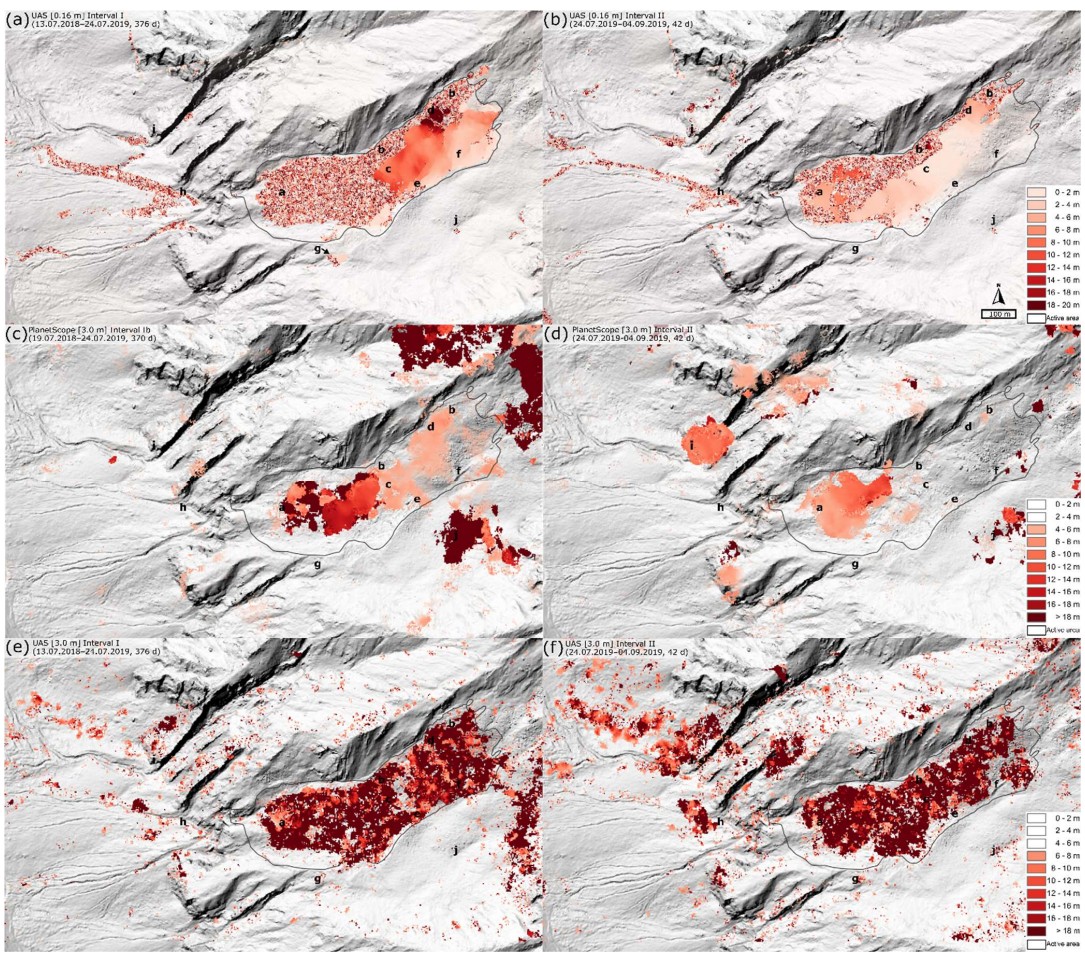

**Figure 5** Results of DIC total displacement of UAS for **(a)** and **(b)** at 0.16 m resolution, **(e)** and **(f)** downsampled to 3 m resolution, and PlanetScope **(c)** and **(d)** at 3 m resolution. Time intervals for UAS image pairs **(a)** and **(e)** are I (13.07.2018–24.07.2019, 376 d), **(b)** and **(e)** II (24.07.2019–04.09.2019, 42 d), for PlanetScope **(c)** Ib (19.07.2018–24.07.2019, 370 d) and **(d)** II (24.07.2019–04.09.2019, 42 d). Solid black line represents the boundary of the active landslide based on field mapping. Background: hillshade of Lidar DEM, 1 m resolution (© SAGIS).

Figure 5a and Fig. 5b show the displacement rates derived from UAS orthophotos at 0.16 m resolution for time intervals I and II (see Table 6). Apart from several minor displacement patches, no motion is visible outside the active body in either period. Time interval I (376 d) (Fig. 5a) shows mean displacement values from 6 to 14 m for a coherent area in the eastern half of the lobe from the centre (*c*) to the eastern boundary of the active area. The highest displacement rates (up to 20 m) are observed within small high–velocity clusters in the northwest section (*d*). Lower velocities occur along the southern boundary (*e, f*), ranging from zero to 6 m with smooth transitions. Ambiguous, small–scale patterns with highly variable displacement rates are present in the western half (*a*) and along the northern boundary (*b*). No motion is detected along the western fringe (i.e. at the landslide head) which is 20 m in width. South of the landslide (*g*) there is a small patch of minor displacement with continuous (up to 3.5 m) and ambiguous signals. Furthermore, we observed small–scale patterns of ambiguous signals in the east (*j*) and in the west of the active area in the drainage channels (*h, i*).

Time interval II (42 d) (Fig. 5b) shows great similarity to time interval I with ambiguous signals in the same areas such as the drainage channels (*h, i*) and within the western half of the active area (*b*). In contrast to interval I (Fig. 5a), within the active area a homogenous higher velocity patch (up to 6 m) near the landslide head is evident (*a*). In the eastern half large homogenous





patches extend from the landslide centre (*c*) to the root zone (*d*) showing coherent displacement values of zero to 4 m. During this shorter time interval II, no displacement is detected along the south eastern boundary (*e*) and for large parts of the root zone (*f*) previously covered in I. Similar to I, the landslide head has a 20 m rim free of signal (also see Fig. 6 *x*, y). In the central part of the lobe (*c*) displacement rates are significantly reduced.

Figure 5c and Fig. 5d demonstrate total displacement for similar time intervals to UAS (see Table 3 and Fig. 4). For interval Ib

(370 d) (Fig. 5c) wide fringes with no motion were detected around an actively moving core area, which consists of small–scale clusters with variable displacement in the western part, coherent high velocities in the middle, and coherent low velocities east of this core area. Outside the landslide, northeast and immediately south (*j*), high–velocity patches are observed.

In interval II (42 d) (Fig. 5d) the detected displacement is restricted to the western half of the landslide (*a*) and shows the same significant fringes with no motion as in I. Compared to interval I the motion pattern of this core area is more homogeneous

with increasing displacement towards the east. Outside the active area several patches show medium to high displacement, the largest of which is located 300 m northwest of the landslide (*i*).

In Fig. 5e and Fig. 5f displacement results of UAS downsampled to 3 m are compared to PlanetScope at 3 m for both time intervals, I and II. Overall, the results demonstrate high rates (~ 18–20 m) across the entire landslide interrupted by scattered speckles of low to medium displacement (*a, b, e*). No motion was present in a fringe zone along the landslide front (west

boundary), similar to results in Fig. 5a and Fig. 5b. In general, the displacement patterns are less smooth than at 0.16 m input resolution. Outside the landslide significant displacements exist at the eastern image border (Fig. 5e) and towards the west (*h, i*) (Fig. 5f). In comparison, displacement rates derived from PlanetScope cover in large parts the active area for Ib (Fig. 5c); however, for II only the core area of the landslide shows displacement. In both results the core areas of the landslide are surrounded by wide fringes with no data.

**5.2. Single Block Tracking**

Figure 6 illustrates the displacement rates derived from the UAS data at high resolution (0.16 m) for interval II (42 d). UAS orthoimages were used to manually measure single block displacement for 36 clearly identifiable boulders on the landslide surface. Block displacements of 1 m are visible in the eastern part (*f*), whereas DIC does not reveal any displacement below 1 m. Boulder tracks longer than 2 m in the central and western part of the landslide are reflected by DIC–derived displacement

values. Near the front a 6 m displacement of one block (*a*) is represented in the DIC result. The highest values (6 m, 10 m, 16 m) were observed in regions where DIC delivered ambiguous, small–scale patterns of highly variable displacements.

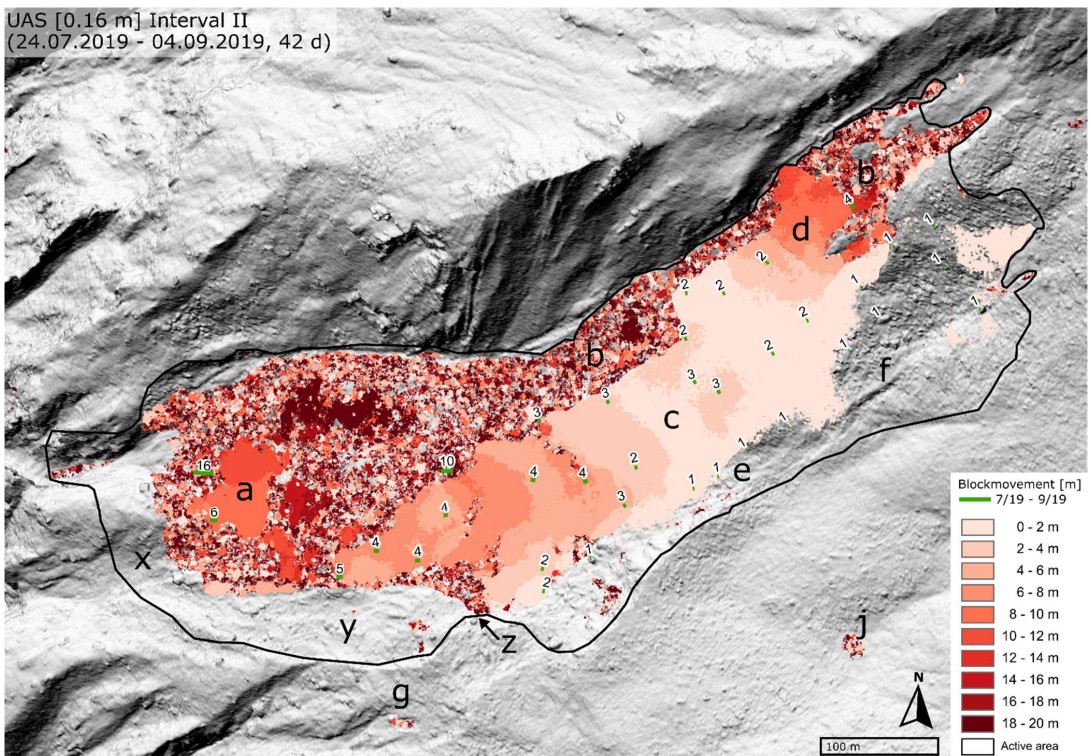

**Figure 6** Displacement derived from UAS data at 0.16 m resolution for interval II (24.07.2019–04.09.2019, 42 d) combined with boulder trajectories (in metres) manually measured in the UAS orthophotos in the same time period. Background: UAS hillshade, 24.07.2019
(0.08 m), orientation -3° from north.

### 5.3. Time required for collection, processing and evaluation

In Sect. 2 we introduced a novel concept to extend lead time, consisting of three phases within the warning time window (see Fig. 1Figure 1). This concept is based on DIC results, thus every step comprised in each phase has been previously undertaken. On this basis, knowledge of required time for a further process iteration of the three phases is given.

Time required for collection, processing and evaluation of UAS and PlanetScope data are estimated and summed in Fig. 7. PlanetLabs specifies 12 hours from image acquisition to the provision in the data hub, which includes to a large amount data pre–processing (Planet Labs, 2020b). Adding two hours for the selection, order and download process, we assume that time required for the collection phase is approximately the same for both sensors, with 14 hours for PlanetScope and 12 hours for UAS. With regard to the time needed for the processing phase, the sensors differ with UAS requiring 17 hours and PlanetScope
five hours. Time for the evaluation phase is estimated to be about two hours. In sum, $t_{warning}$ for UAS is approximately 31 hours compared to 21 hours for PlanetScope.



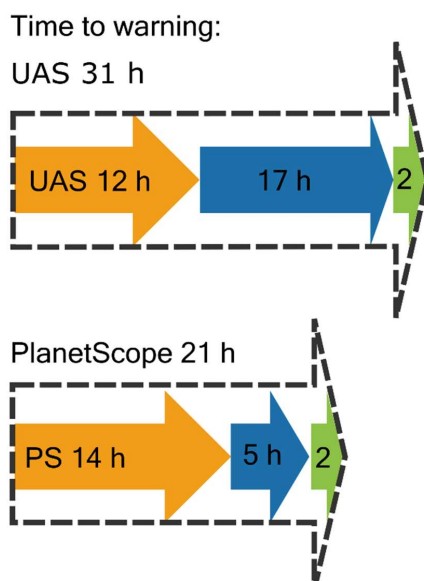

**Figure 7** Time to warning is composed of three phases: time to collect, to process and to deliver. Time to warning (subsequent to acceleration) is 21 h for PlanetScope and 31 h for UAS. Thus, any hazard process that takes longer than 21/31 h to prepare the release and impact can be forecasted.


## 6 Discussion

To systematically analyse the predictive power of the UAS and PlanetScope data, we will (i) evaluate error sources and output performance, (ii) assess obtainable temporal and spatial resolution and (iii) derive a systemic estimate of the minimum obtainable warning times.

### 6.1. Error sources and output performance

To evaluate error sources and output performance, we compared results of digital image correlation results from optical data with (i) mapped mass movement boundary, (ii) visual block tracking for UAS and (iii) 3 m downsampled UAS orthophotos. The approximately one year evaluation period encompassed all seasons, hence freezing/thawing conditions and a wide range of meteorological influences, e.g. thunderstorms and heavy rainfall, are included. The two investigated time intervals are I/Ib

and II, covering 376/370 days and 42 days (typical high–alpine summer season), respectively (Fig. 4). Interval II exclusively covers (high–alpine) summer conditions, with negligible to no contribution from freezing conditions. As these inclusion periods are inconsistent, the amount of total displacement cannot be directly compared; however the relative motion patterns can be. Accordingly, we can confirm the suggested parameter settings of earlier studies on window sizes, steps and robustness iterations (Ayoub et al., 2009; Bickel et al., 2018).

In terms of the mass movement boundary, the total displacement derived from the DIC of the UAS data generally matches the field–mapped landslide boundary for both intervals (I, II) (Fig. 5a, b), and is supported by the absence of significant noise outside the AoI. Mapped boulder trajectories for interval II (see Fig. 6) are consistent with the calculated total displacement and thus confirm COSI–Corr as a reliable DIC tool to derive ground motion for this study site and UAS orthophotos as suitable input data. Nevertheless, there are several areas with ambiguous signals. Leprince (2008) describes snow cover, vegetation

cover and alluvial processes, among others, as potential explanations for these decorrelations. In our study, the decorrelated areas include to a large degree the landslide head (*a*), the drainage channel (*h*) (Fig. 5a, b), a larger patch south of the active area boundary (*g*) (Fig. 5a), and some smaller ones in little depressions (*g*) (Fig. 5a) and (*j*) (Fig. 5a, b). Most patches are



identified as snow fields in the orthophotos and the noise results from decorrelation. In Fig. 5a, the large southern patch (*g*)
shows clear displacement values for the rear part and decorrelation for the front region resulting from significant temporal

changes within an image pair, limiting the ability to measure ground displacements. The decorrelation in the drainage channel
(*h*) could stem from massive changes in pixel values, similar to the decorrelation on the basis of alluvial processes, as described
by Leprince et al. (2007). Decorrelations in the areas with the fastest ground motions also lead to high pixel changes (Stumpf
et al., 2016). These are observable in the active landslide area within the lobe, where large areas of decorrelation may be
explained by high displacements in the leading part (*a*) with redetected, hence correlated pixels in the trailing part (*c, d, e, f*).

These findings can be transferred to the landslide interior area (*a, b*), the frontal western regions and the northern margin (*b*).
The observation is confirmed by geomorphological mapping and measured boulder block trajectories from the orthophotos
(Fig. 6). Several patches of correlation (*c, f*) with corresponding boulder trajectories up to 4 m (*d*) are detected in the rear part.
A correlated patch with a 16 m trajectory (*a*) is in flow direction behind the foremost boulder. In this case the method was able
to capture the displacement partially as the distinct boulder block supported the detection, hence correlation. This allows us to

conclude that displacements exceeding approximately 10 m for the calculated time period, thus 63 pixels or more at a resolution
of 0.16 m, are definitely outside of a possible correlation and no pixel matching is possible. With a correlation window smaller
than the displacement, the algorithm is not able to capture the displacement (Stumpf et al., 2016). Field observations provide
evidence that the surface alters due to the high mobility and rotational behaviour of some boulder blocks, which leads to
changed pixel values and spectral characteristics. Similar results were observed by Lucieer et al. (2014), who described a loss

of recognisable surface patterns if revolving and rotational displacements occur, causing decorrelation and a noise as output.
These results show that with COSI–Corr and UAS orthophotos of 0.16 m, it is possible to detect the total displacement of the
landslide in both extent and internal process behaviour even in this steep, heterogeneous terrain. Nevertheless, high
displacement rates and rotational surface behaviour in the cirque limit the DIC method. A decrease of the time interval for this
particular highly mobile study site would likely reveal an enhanced correlation since for shorter time periods the total

displacement decreases, and surface changes are reduced, which can be controlled by shortening the temporal baseline.

### 6.2. Comparison of temporal and spatial resolution

We compared the COSI–Corr total displacement results of PlanetScope (Ib and II, Fig. 5c, d) and UAS images (I and II,
Fig. 5a, b) for the same time periods at different spatial resolutions (see Table 6). For the PlanetScope DIC result the main part
of the landslide is detected, and its area is generally consistent with the results of the UAS DIC, which is additionally confirmed

by boulder trajectories. The frontal part (*a*) reveals correlation signals (I and II); while for the same time intervals and parts, the
UAS DIC results show a decorrelation (Ib and II). The correlation is likely to be attributable to the coarser spatial resolution
of 3 m input data, hence a smaller number of pixels to be captured at this site with the DIC method. Similar texture of rock
clast surfaces could lead to false positives resulting in correlation as patches appear similar in matching windows. However,
in contrast to the UAS result (Fig. 5a, b), the outcome on a large scale fails to detect the entire actual active area (*b*), (*f*) as well

as its internal motion behaviour. Nevertheless, for the visualisation and analysis of the PlanetScope results, the range of total
displacements had to be restricted to values equal to and greater than 4 m due to noise and outliers over large areas, as applied
and described by Bontemps et al. (2018). Even then, noise and several misrepresented displacement patches are observed for
(*i, j* and in the northeast image corner (Fig. 5). We can identify several reasons for these large clusters of high motion values.
Massive cloud and snow coverage hampered both first images of interval Ib (19.07.2018) (Fig. 5c) and II (24.07.2019)

(Fig. 5d), leading to a 20 m fringe of false displacements in the north–eastern part of the image. Minor snow fields could
explain the big cluster of incorrect displacement southeast of the lobe (*j*); nonetheless, in the satellite image they are smaller
than the resulting DIC displacement. High cloud coverage in two input images with large areas of white pixels may exert an
influence leading to high gains due to sensor saturation (Leprince, 2008). Illumination changes in interval II (Fig. 5d) may
cause unrealistic displacements outside the boundary with slightly darker colours due to shadows in the first satellite image





(24.07.2019) and large parts within the second image (04.09.2019) are also in the shade. A comparison of the acquisition times and true sun zenith, e.g. for the second image, reveals a difference of 01:34 h between the image acquisition at 11:36 LT (local time) and the true local solar time at 13:10 LT. As the study site is located in a high–alpine terrain with a west facing cirque, at this time of day there are shadows of considerable length which have a significant influence on the result of digital image correlations. One clear advantage of the UAS images is that their acquisition is plannable according to the best illumination

conditions with the sun at its zenith. Moreover, the UAS flight path as well as the system itself remained the same for all three acquisitions, while PlanetScope employs various satellites.

Despite similar input resolutions and time intervals (Ib vs. I and II vs. II, see Table 3) with different sensors (UAS, PlanetScope), considerably divergent DIC outputs (Fig. 5c vs. e, d vs. f) are returned. To a large degree the active ground motion inside the mapped landslide boundary is represented by the 3 m UAS DIC result, while the same fringe remains free

of signal for both UAS DIC results at different input resolutions (Fig. 5Figure 5a, b vs. Fig. 5e, f). This similarity with overall good agreement indicates that the displacement is restricted to a smaller area than the previously demarcated boundary, based on our field investigations. The satellite image detects large parts of the main active core area but widths of 50–80 m from the boundary show no displacement. False displacement is indicated for a cluster outside of the boundary to the image border in the east for UAS interval I (Fig. 5e) and in the north western area (*h*, *i*) for interval II (Fig. 5d). Apart from these false signals,

there is minor noise compared to false large clusters of high displacement within the PlanetScope result interval I for (*j*) and northeast image corner (Fig. 5c) and interval II (*i*) (Fig. 5d).

However, two striking differences with correlation/decorrelation and ground motion values are observed for the two UAS input resolutions; the coarser resolution of 3 m returns a correlation signal with values typically exceeding 18 m of displacement as the value range is extended, due to previous high factor downsampling. Measured ground motion of block

tracking and PlanetScope results indicate and support existing high ground motions. This observation might be the explanation for the observed decorrelation at the finer resolution of 0.16 m for the landslide head. For this reason, the previous assumption using a shorter time interval leading to improved detection of inherent process behaviour (see Sect. 6.1.), can be complemented with a coarser resolution showing a clear improvement in the form of better correlations and returned signals. Generally, with high resolution images, such as UAS, we recommend first calculating displacements based on a coarser input resolution (1–

3 m) to examine the overall situation and detect changes, and second to calculate displacements at a finer resolution in order to focus on relevant details of the AoI. With regard to PlanetScope data, a 3 m resolution seems to be in a good spatial range to assess ground displacements even of this steep and heterogeneous study site with its high motion. Nonetheless, constraints such as illumination due to early daytime acquisitions leading to shadows, meteorological influences by clouds, cloud shadows and snow decrease the quality of the satellite images and reduce their applicability. Sensor saturation, shadow length, size and

direction as well as changes in snow, cloud or vegetation cover impose limitations (Delacourt et al., 2007; Leprince et al., 2008) and accord with our observations. The authors identify additional limitations such as radiometric noise, sensor aliasing, man–made changes and co–registration errors (ibid.). All these limitations have a negative impact on the input image, which leads to impaired DIC calculations and results, and (partially or wholly) inaccurate analysis of the displacement. These might have played a role in our results. In our experience, the usability of the DIC result may be influenced by the input image

quality. This restricts the application of PlanetScope images to a certain degree. They can be employed as input data to detect displacements, but as there are in the present setting too many signals of false–positive displacements, which can solely be discarded on the basis of field evidence, this data is currently of limited use. It should be handled with caution, and combining it with complementary data and ground truth is recommended.

### 6.3. Estimating time to warning

Early warning is essentially defined as being earlier than the event and thus puts high external time constraints on observation and decision. The time window between the detection of an accelerating movement preparing for final failure and the final





failure itself is determined by the environment. Therefore, two sensors with the highest available spatiotemporal resolution were evaluated and compared with regard to their applicability to the early warning of landslides. We made rough assumptions and assessed the time needed for the phases of time (i) to collect, (ii) to process, and (iii) to evaluate relevant data (summarised

in the time to warning window, see Fig. 7).

Despite different underlying technologies the time required for the collection phase is approximately the same for both sensors. For UAS, we estimated about 12 hours under ideal circumstances, while for PlanetScope 12 hours (Planet Labs, 2020b) plus two hours for image selection, download and initial analysis, adding up to 14 hours in total (see Sect. 5.3.). In the second phase, time to process, deriving orthophotos from raw UAS images is time consuming. The subsequent DIC calculations

demand significantly more processing time for the UAS images than for lower resolution PlanetScope images. The final phase, time to deliver, takes about two hours for each sensor. In our case study, the estimated time to warning ($t_{warning}$) was 10 h longer for the UAS approach (31 h) in comparison to the Planet Scope approach (21 h). These time calculations are based on ideal environmental conditions and data availability. Assuming good conditions exist to conduct the UAS flight and no constraints limit the utilisation of satellite images, in theory a daily deployment is possible. In reality, unfavourable weather conditions,

cloud and snow cover as well as limited data availability will increase the actual $t_{warning}$ significantly. From the available images in the Planet Data hub (besides other exclusions) meteorological influences reduced for April–October 2019 the usability by 14.5 % and 7 % for cloud cover and snow cover, respectively (Table 2). The flexibility of a UAS can serve as a practical remote sensing tool for the investigation of ground motion behaviour in a spatiotemporal context. Nonetheless, weather influences can make a UAS flight impossible or impractical as the result might be useless. Depending on the level of

illumination, the same may apply for satellite images. Regardless of any meteorological constraints, the promised daily availability by PlanetScope is unrealistic, due to data gaps and provider issues; our study showed that for the Sattelkar from April to October 2019 only 11 % of the captured images during this time were usable. In time–critical early warning scenarios, when time is running out, all available even partly usable images will be utilised and fieldwork may be conducted, even if the prevailing conditions are suboptimal but will increase data availability. The comparison of two selected remote sensing options

demonstrates that the comprehensive knowledge on the available remote sensing data sources and their respective time requirements can substantially reduce the time to warning ($t_{warning}$) and to extend the lead time ($t_{lead}$).

Significant observations of the temporal evolution of historic landslides are presented in Table 7 and described below. These include (i) the Preonzo rock slope failure, CH (Sättele et al., 2016; Loew et al., 2017), (ii) the Vajont rock slide, ITA (Petley and Petley, 2006) and (iii) the Sattelkar complex slide, AUT (Anker et al., 2016). These landslides have specific evolution

histories, e.g. early observed crack developments, increased movement and minor events like Preonzo (2002 and 2010) (Sättele et al., 2016); Sattelkar, with large volume mass wasting processes since 2005 and a debris slide event in 2014 (see Sect. 3 Study Site) (Anker et al., 2016); and Vajont, with ductile failures in 1960 and 1962 and a transition from ductile to brittle behaviour in 1963 (Petley and Petley, 2006; Barla and Paronuzzi, 2013).

**Table 7** Relevant dates for historic failures of Vajont (ITA), Preonzo (CH) and Sattelkar (AUT). Time period in italics–bold used for Fig. 9. Time intervals in days (~ for rough estimations) and years in square brackets; sum of days based on the first day of the month, if only month as reference is available from literature (Petley and Petley, 2006; Anker et al., 2016; Sättele et al., 2016; Loew et al., 2017). Further explanation below.

| Vajont [days, yrs] | | | | Preonzo [days, yrs] | | | | | | Sattelkar [days, yrs] | | | | |
|---|---|---|---|---|---|---|---|---|---|---|---|---|---|---|
| 10/1960 | large crack | | | 1989 | onset crack | ~730 | | | | | | | | |
| **04.11.1960** | 700.000 m³ block detachment | | | 1991 | increasing movement | [2] | ~4'138 | | | | | | | |
| 11/1960 | 11 mm/d | | | **05/2002** | minor event | [11] | | ~2'922 | | | | | | |
| 04/1963 | ductile to brittle transformation | 153 | | **05/2010** | minor event | | [8] | | ~731 | 01/2003 | enhanced dynamic mass wasting | ~731 | | |
| 09/1963 | 35 mm/d | [-] | ~30 | 01.05.2012 | critical displacement | | [2] | 15 | | 01/2005 | begin of active ground motion | [2] | ~3'498 | ~4'229 |
| **09.10.1963** | **260 mio. m³ event** | [-] | | 15.05.2012 | 300.000 m³ event | | [-] | | | **31.07.2014** | **170.000 m³ event** | [9] | | *[11]* |

Figure 9 is the extension of our concept (see Sect. 1, Fig. 1Figure 1) systematically supplemented with our estimated time to warning (UAS, PlanetScope), and compared to the few data series predating larger slope failures.





Following a significant acceleration, the forecasting window is opened and $t_{warning}$ starts, which is composed of phases (i) time to collect, (ii) time to process and (iii) time to evaluate. To ascertain a significant acceleration one further observation is required. Hence, one complete cycle of the three phases, previous analyses and processing iterations are given. Our analysis

showed that UAS and Planet Scope can approach times as short as 31/21 h, as a result $t_{lead}$ is increased and so is $t_{react}$.

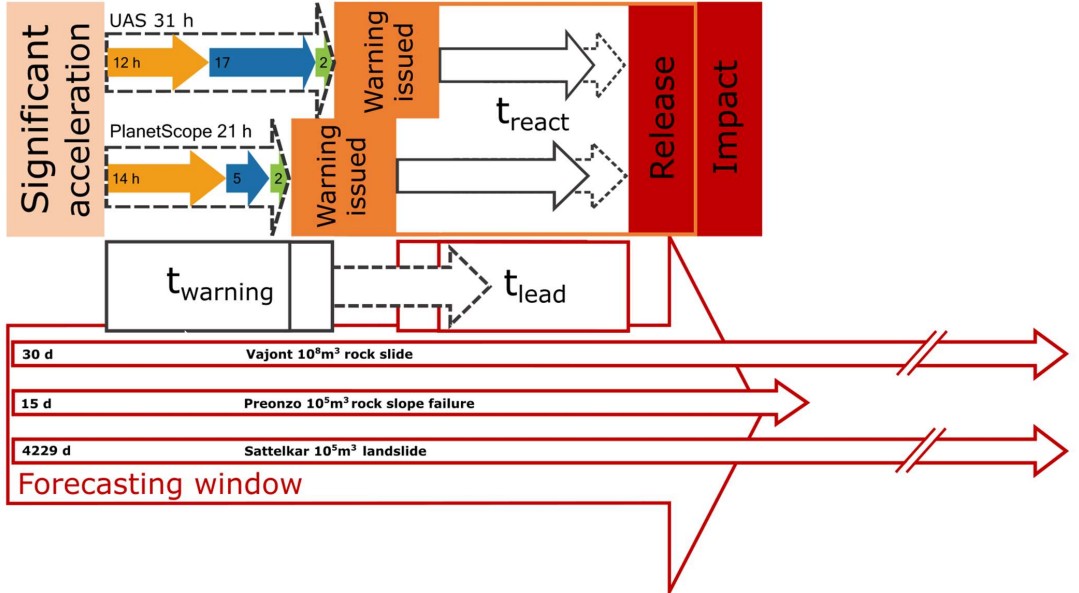

**Figure 8** Conceptual approach with estimated $t_{warning}$ for UAS and PlanetScope. Phases of collection, processing and evaluation (indicated as arrows of relative length in orange, blue and green, respectively) (see phases in Fig. 1 and Fig. 7) with their total duration time (grey

dashed arrows). In $t_{warning}$, one additional observation requires in sum 31 for UAS and 21 h for PlanetScope data. Above, major landslides are compared from the onset or displacement detection (solid line) (Petley and Petley, 2006; Anker et al., 2016; Sättele et al., 2016).

Assuming both sensors reliably estimate ground motion, solely based on their time requirement, this concept was applied to the temporal development of historic landslide events, thus from measured increased displacements and/or massive accelerations to the final event (Table 7). On this basis we simplified the graph and what we defined as "significant

acceleration" using dates of observations such as increased crack opening (Vajont), critical displacement (Preonzo) and the beginning of active ground motion (Sattelkar). Therefore, the opening of $t_{warning}$ and forecasting window are concrete observations of the particular site, independent of any intensity described by the corresponding authors and allows more freedom for temporal evaluations without going into details.

For the Preonzo case, the entire 2012 spring period was characterised by high displacement rates. We defined the first of May

2012, when geologists operating the warning system informed local authorities and assembled a crisis team, as the onset or 'increased movement' and the 15.05.2012 with 300 000 m³ as the impact (Sättele et al., 2016), in total approximately 15 days. For Vajont, the 1/velocity plot by Petley and Petley (2006) (based on data from Semenza and Ghirotti (2000)) shows an increase in movement at about day 60 along with a transition from a linear to an asymptotic trend at approximately day 30, defined as a transition from ductile to brittle. Therefore, we assumed 30 days of forecasting window for $t_{warning}$ and $t_{lead}$ until

the impact of the hazardous event on 09.10.1963. For the Sattelkar site, the observed mass displacement increase is presumed to have started in 2005 with the 170 000 m³ debris flow event on 31.07.2014 as the impact, thus about 3 498 days (Anker et al., 2016).

Even for the Preonzo event, with its short forecasting window of 15 days, the ground motion assessment based on the evaluated optical remote sensing images, would have been possible under the assumption of reasonably good UAS flying conditions and





the provision of usable PlanetScope images. For $t_{warning}$ there is enough temporal leeway to repeat at least three to four successive measurements comprising the three phases. However, as single accelerations are possible in very short time intervals of less than two days, it is impossible to capture these accelerations by means of optical remote sensing methods, given a time requirement of 31 hours for UAS and 21 hours for PlanetScope. Nevertheless, this comparison shows that for larger and long–preparing slope failures the technical $t_{warning}$ may well be shorter than the forecasting window starting at the

time at which the process became predictable.

### 7 Conclusions and outlook

This paper presents an innovative concept to compare the lead time for landslide early warning, utilising optical remote sensing systems. We tested this temporal concept by applying UAS and PlanetScope images of temporal proximity as these are currently the sensors with the best spatiotemporal resolution. We assessed the sensors' capability to identify hot spots and to

recognise behaviour by delineating ground motion employing digital image correlation (DIC). In so doing, knowing the necessary processing time enabled us to estimate the time requirement and finally to incorporate it into the concept to evaluate sensors with regard to ongoing landslide processes of the Sattelkar as well as historic landslide events.

This paper presents an innovative concept to compare the lead time for landslide early warning, utilising optical remote sensing

Our findings derived from DIC for this high–alpine case study show that high resolution UAS data (0.16 m) can be employed to identify and demarcate the main landslide process and reveal its heterogeneous motion behaviour as confirmed by single

block tracking. Thus, validated total displacement ranges from 1–4 m and up to 14 m for 42 days. PlanetScope Ortho Scenes (3 m) can detect the displacement of the landslide central core, however, cannot accurately resolve its extent and internal behaviour. The signal–to–noise ratio, including multiple false–positive displacements, complicates the detection of hotspots at least in this very steep and heterogeneous alpine terrain.

With regard to the temporal aspect for early warning purposes, PlanetScope satellite images require less time compared to

UAS for the time phases of collection, processing and analysing. As a consequence, when time is of the essence, the UAS acquisition cannot compete with the high frequency of PlanetScope daily revisit rates. In general, both are limited in their use as they are passive optical sensors dependent on favourable weather conditions. Nevertheless, with a realistic 10 % of usable data for our study site, PlanetScope cannot provide daily data as promised.

To conclude, in methodological terms DIC is a reliable tool to derive total displacement of gravitational mass movements even

for steep terrain. Given the high reliability of UAS data, its temporal resolution is the key in future attempts to overcome decorrelation due to high ground motions. In addition, a slightly coarser resolution reduces the time needed for total processing, enhances correlation while maintaining spatial accuracy and reliability. PlanetScope is especially interesting as a complementary sensor when UAS employment is restricted e.g. inaccessible and/or dangerous sites or for areas too extensive to be covered. For continuous monitoring and early warning, the warning time window could be shortened by on–site drone

ports with autonomous acquisition flights and automatic processing. Our systematic evaluation of the sensor potency can be applied and transferred to other optical remote sensing sensors, the same is true for our conceptual approach optimising extending the lead time. Future studies should focus on the applicability of complementary optical data to confirm the detection of landslide displacement and adjust UAS output resolution as this significantly increases the validity of DIC internal ground motion behaviour.




**Data availability**

PlanetScope data are not openly available as and PlanetLabs Inc. is a commercial company. However, scientific access schemes to these data exist.

**Author contribution**

Doris Hermle developed the study together with Markus Keuschnig and Michael Krautblatter, analysed the data and wrote the paper. Markus Keuschnig and Michael Krautblatter supported the writing and editing of the paper. Ingo Hartmeyer provided critical proof reading with valuable suggestions. Robert Delleske is responsible for UAS flight campaigns and processing the images.

**Competing interests**

The authors declare that they have no conflict of interest.

**Acknowledgements**

This work is supported by a scholarship of the Hanns–Seidel–Foundation. The authors are grateful to PlanetLabs for their cubesat data via Planet's Education and Research Program. We thank Tobias Koch for the support in fine co–registering satellite images.

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
