# Peer review of "Challenging the timely prediction of landside early warning systems with multispectral remote sensing: a novel conceptual approach tested in the Sattelkar, Austria"

_Natural Hazards and Earth System Sciences, 2021_

## Author Comment (AC1)

**Letter of response to comment on nhess-2021-18**

Dear Jan Blöthe,

We thank you and appreciate your valuable comments on our manuscript. Your feedback has helped us to improve our work and pointed to areas which were ambiguous and therefore needed clarification.

Please find below the following colour coding for the review and your comments in black; our responses to the review are in blue and the changes made to the manuscript are in green.

**General comments**

**A) Description of digital image correlation method and error assessment**

In my view, digital image correlation is not a trivial method and deserves a more detailed description in section 4.3. Especially because the conceptual approach presented here grounds on the detection of significant movement (or even acceleration) from optical imagery, the authors should elaborate the exact processing steps and include a detailed accuracy assessment. This can easily be achieved by:

- The quantification of a level of detection between images, i.e. the residual mismatch of stable surfaces outside the landslide between consecutive images after image correlation, beyond which significant displacement can be detected with a given confidence.
- Excluding spurious matching results (displacement vectors) on the basis of a correlation threshold.

The description of section 4.3., Data Acquisition and Processing, has been modified by adding more details.

The attached Online Supporting Material (OSM) contains the variety of results which show our approach to selecting the appropriate combination of UAS input data (orthophotos, DSM and hillshade derivates) and displacement vectors (see OSM Figs. 7, 8 and 9). In addition, signal to noise results and volume calculations are provided (see OSM Figs. 3, 5, 8, 9 and 11, 12). The distribution of GCPs combined with DIC total displacement results of UAS are also presented (see OSM Fig. 1 and 4).

In terms of the selection of appropriate parameter settings, we decided to use:

- for a step size of one, as larger step sizes smoothed the velocity pattern, did not obviously improve the matching while decreasing the spatial resolution. Computation time would decrease if larger step sizes are employed.
- UAS 128 x 32, as an initial window of 256 returned a general decrease in velocity. Furthermore, the smaller initial window of 64 matching was only partially successful with very low velocities. The final window size is important to detect small scale features. If set too large, features could be smoothed out. In our case there were no distinct differences, which is why we selected the smaller final window option: to necessary small scale features.

However a detailed accuracy assessment requires comparable data which is not available such as in the verification process of DEM production based on stable surfaces. Therefore, we

added the signal to noise results as you requested in the OSM. An accuracy assessment similar to Travelletti et al. (2012) having GCPs within the active landslide cannot be conducted, as in contrast our GCPs are located on stable positions outside of the active landslide (see OSM Fig. 1 and 4). Our approach to this study is to compare manual block tracking with the calculated velocities from DIC as part of the data evaluation.

**B) Result of image correlation**

As stated above, digital image correlation and the extraction of displacement from correlated imagery is not a trivial task and many pitfalls can lead to spurious results (the authors term these decorrelated). I will outline my doubts regarding the validity of the obtained displacement values referring to Fig. 5, but have given many detailed comments on the respective text positions in the specific comments below. In large areas, the image correlation returns areas that are "decorrelated", such as the western part of the landslide in (a) and (b), but also positions in (e) and (f) are affected by this. In my experience, such a pattern indicates that matching between images did not work, which should be visible by adjacent vectors having very different magnitudes and directions. Furthermore, the patchy nature of displacement values in the western part of (c) is very surprising. Here very high total displacement of ~18 m is located in the vicinity of displacement on the order of 4-8 m. From an image matching procedure, I would expect a rather smooth picture here, such as in (d). But also from a geomorphic perspective, I am unsure how this pattern could be explained by a natural process. Finally, the results obtained from the downsampled UAS DEMs predominantly show high rates (16-18 m) that are interrupted by areas of no movement or very slow movement. My impression would be that these results are least reliable, because a) they show a completely different picture as (a) and (b), while being computed with the same data (just a different resolution), b) the displacement values are nearly the same for two very different time intervals (e = 376 days, f = 42 days), c) they are not matching the values obtained from manually tracking boulders (again, based on the same data), and d) I am unsure if such a pattern can be produced by a natural process.

Having outlined my reservations regarding the image correlation results, let me suggest a couple of strategies to improve the results:

- Use a hillshade not a DEM for tracking (not clear if this was done)
  Originally we used UAS orthoimages. Please see the OSM Fig. 8 for calculations using DSM and OSM Fig. 9 using hillshades.
- Resample the DEM to a slightly coarser resolution (0.5 m?)
  We have tried a 0.5 m resolution for the UAS orthophotos with different parameter settings showing overall better matching with still some decorrelation. However, with this input resolution and the best suited parameter settings of 128 x 32 the extent is already decreased in its size to a smaller displacement area.
- Try a different software for image correlation, there are many and all have their advantages and disadvantages
  This was done with DIC–FFT and IMCOOR (please see OSM Fig. 10 for results of DIC–FFT).

- Have a detailed look into the correlation coefficients and the bearings of the displacement vectors and exclude spurious results.
  Please see OSM Fig. 3 (b) and (h) with displacement vectors and signal–to–noise maps in the OSM Fig. 3 (c), (i) and (d), (j), Fig. 5 (f) and (i) and a cross profile cutting the DIC total displacement for both intervals I and II, Fig. 6.

Yes, indeed a mismatch of the initial and final search windows, i.e. a decorrelation, is visible for many areas but especially obvious in the western part of our DIC results. The current literature states among others that there is an upper limit regarding velocities of ground motion (Delacourt et al. 2007; Travelletti et al. 2012). In this area very high motion clusters of this complex landslide exhibit debris slide characteristics. We observed that acceleration of the landslide body takes place here. In contrast, in the eastern part of our DIC results, there are correlated areas and smooth motion patterns indicating that matching took place and the method was successful with the applied parameter settings.

Additionally, in our case, the terrain surface is altered rapidly; big blocks with edge lengths of up to 10 m rotate and cause significant surface changes, which could be a further reason for decorrelation (see OSM Fig. 9 for results of DIC–FFT) (Lewis 2001; Stumpf et al. 2018). The geomorphic causes for the observed acceleration are unknown but could be related to permafrost degradation and increased infiltration of rain- and meltwater.

In the OSM we support the result from DIC with the corresponding displacement vectors (OSM Fig. 3 (b) and (h)).
With regard to the 3 m downsampled UAS orthophotos we are aware that these results are less trustworthy in terms of delineated velocities. Here, our purpose was to compare two different sensors in order to see how accurate PlanetScope data are for high alpine displacement calculations. Please see here our comment further below.

- L22/23: While this is certainly true, the authors should elaborate in the introduction that events instantaneously triggered by earthquakes or heavy precipitation are beyond what their proposed framework can deliver an early warning for. The necessity of gathering and evaluating data prior to issuing a warning limits the analysis to mass movements that indeed show a pre-failure acceleration on the order of days.
  Thank you for highlighting this. We totally agree that this has to be mentioned in the beginning to complement our explanations in the discussion, L561/562.
  L31: This definition of an early warning system (EWS) contains a time component but includes no exact time scale reference. ~~'Early' suggests that events are detected before harm or damage occurs and thus stands in contrast to events which are only detected once they have begun (e.g. snow avalanches). Thus, it is necessary to know sensor capabilities and limitations for pre event mass movement observations (Desrues et al., 2019). The success of a warning requires that information is provided with enough lead time for decisions on reactions and counter measures (Grasso, 2014).~~ The success of an EWS therefore requires measurable pre-failure motion (or slow transport velocities) to allow for sufficient lead time for decisions on reactions and counter measures (Grasso, 2014). In this regard, knowledge on sensor capabilities and limitations is essential, as it determines which rates and magnitudes of pre-failure motion can potentially be identified (Desrues et al., 2019). Our proposed framework refers to mass movements with significant pre-failure motion operating over a sufficient time periods and thus excludes instantaneous events triggered by processes such as heavy rainfalls or earthquakes.

- L25/26: Is this really just attributable to the warming of the climate?
  To the best of our understanding and following Gariano and Guzzetti in their review (2016) the global climate warming directly and indirectly impacts natural and human induced factors which can again directly or indirectly condition landslide activity, abundance and frequency of events. Other reasons for landslide triggers are included in L22/23, earthquakes, rainfall events and human interaction.

- L47/50: I would think that also the rate of landslide movement defines whether or not it can be detected by optical imagery.
  Thank you for pointing out that detection is not restricted to sensor characteristics. This is very important to say, of course.

  Recently, the spatial and temporal resolution of optical satellite imagery has significantly improved  (Scaioni et al., 2014) and has allowed substantial advances in the definition of displacement rates and acceleration thresholds to approach requirements for early warning purposes. This is essential since spatial and temporal resolution determines whether landslide monitoring is possible with the detection  of displacement rates and the approximation  of acceleration thresholds, which both are lacking if information is based solely on post–event studies (Reid et al., 2008; Calvello, 2017).

- L79/80: This is the maximum revisit time at the equator, right? For the study area shown here, revisit time should be shorter.
  Yes, thank you for mentioning this. We will differentiate here between revisit frequency and repeat frequency, with the latter of importance for coherence.

  One advantage of optical imagery is its temporally dense data (**Fehler! Verweisquelle konnte nicht gefunden werden.**) compared to open data radar systems with sensor repeat frequency every six days and revisit frequency between three days at the equator, about two days over Europe and less than one day at high latitudes (Sentinel–1, ESA).

- L121: What do you mean by "natural developments" and how are these conditioned or different from natural processes?
  Thank you for this comment. We are sorry that this was not specific. We meant the development of natural processes.

  Natural processes and their developments constantly take place independently, thus dictate the technical approaches and methodologies researchers must apply within a certain time period.

- Figure 1: While I like the idea behind this conceptual figure, I would recommend the authors add a time axis and limit the area of "significant acceleration" to a vertical line that coincides with t = 0. In the present form, the conceptual figure contradicts statements in the text, such as "The forecasting window is started […] following significant acceleration […]" (L126), or "Simultaneously with the forecasting window, time to warning (t_warning) starts (grey outline)" (L128/129).
  Thank you, you are right. We changed it to our best understanding of your feedback.

[Figure]

- L133/134: This also does not match what Fig. 1 is showing

  "The lead time is the difference between the forecasting window and the time to warning."

  We want to express that $t_{lead}$ is the rest/remainder of the subtraction as follows

  $$t_{forecasting\ window} - t_{warning} = t_{lead}$$

  Please let us clarify this as it seems to be some sort of misunderstanding here.

  As a suggestion, this could be replaced with L133/134, if you prefer "Lead time is the forecasting window minus the time to warning."

- L139: This also does not match what Fig. 1 is showing. In Fig. 1, tlead < treact.

  *"An imperative for an effective EWS, the required time to take appropriate mitigation and response measures has to be within the lead time interval (tlead) (Pecoraro et al., 2019) with tlead ≥ treact".*

  Please let us try to clarify this: in best case, the lead time is longer than the time needed to take responsive measures and react to the impending event ($t_{react}$), this is indicated by the shorter solid grey arrow. However, if the reaction time is as long as the lead time, see dashed extension of the grey arrow, then it is a coincident ending of both, $t_{react}$ and $t_{lead}$ prior the release and impact.

- L215/127: In theory yes, but as you show later (Tab. 2), the effective revisit time of optical imagery might in fact be very similar.

  Unfortunately, we do not understand what you are referring to in L127.

  L215: Sentinel–1 does have a revisit time of about every second day over Europe.

  However, the repeat frequency for coherence to generate interferograms is every six days.

  This is the shortest possible temporal baseline.

In terms of optical satellite images, yes, this is what the author team finally wants to lead to. PlanetLabs claim to have daily acquisitions and thus can provide daily imagery supply. But upon a closer look the practitioner knows the reality is different. This has to be kept in mind if this kind of data is employed for the purpose of a reliable monitoring and process observation. For this reason, Table 1 and Table 2 have different and contradicting statements, in this case for PlanetScope.

You are right in some way: free satellite images by Sentinel–2 are, at five days, very close to the six days for interferograms by Sentinel–1, given that both sensors are suitable for the given characteristics by the acquisition target (motion velocity, exposition). Apart from open data providers, there are many others providing even sub–daily acquisitions such as WorldView 3/4.

- L242/248: It might be worth mentioning here that on average, only 11% of the images were usable, significantly reducing the theoretical revisit time, as you also outline in the discussion.
  Thank you, indeed this is worth to be mentioned and we changed accordingly.
  In this seven–month period, 43 images (20.1 %) had data voids or did not cover the AoI, thus the overall usability is limited to about 11 %.

- L267/269: Please elaborate how you filtered for "errors of location, shift and spectral colour problems" (are the latter spectral differences between images?).
  We used QGIS software to manually select the satellite images with the reference UAS images at the base and the visual "show/hide" of the satellite slave images on top. Similarly, the application Map Swipe Tool plugin was employed by dragging the slider across the images.
  Spectral colour problems are shifts in the individual r, g and b bands within one single image:

[Figure]

The other shifts which might occur cannot be corrected for. The first time these can be detected is in a GIS software with the visual check previously described:

[Figure]

Thereafter, a second selection (visually with the Map Swipe Tool plugin) from the downloaded images was filtered for errors of location, shift and spectral colour problems which were previously not clearly discernible in the online data hub.

- L281/285: Please specify the accuracy of dGPS coordinates as measured for the GCPs and also include an accuracy information for the DEMs and their derivatives that were produced from UAS surveys.
  The accuracy of dGPS coordinates, which were employed for the processing of UAS data and DEM/orthophoto generation, range between 5 cm horizontally and 10 cm vertically. All UAS model calculations are based on the same dGPS measurements.
  The RMS errors from UAS image processing in Pix4Dmapper range between 4 and 8 cm. If generation reports are necessary, they can be provided on request later (due to current office access difficulties).
  These were repeatedly (1000 measurements/position) registered with the TRIMBLE R5 dGPS and corrected via the baseline data of the Austrian Positioning Service (APOS) provided by the BEV (Bundesamt für Eich– und Vermessungswesen). Horizontal root–mean–squared errors (RMSE) range from 0.05 m to 0.10 m for vertical RMSE. These GCPs were employed for georeferencing and further rectification of all UAS surveys.

- L285/286: Please elaborate how image co-registration was achieved and state here the residual mismatch between co-registered images.
  DIC methods for estimating terrain movements require accurate geo-referencing of consecutive satellite images avoiding falsely detected systematic drifts. Although the investigated satellite sensors are equipped with high–quality geo-localization sensors, subtle deviations in the absolute geo-referencing rates are expected for different acquisition times.
  Therefore, a fine–registration between satellite image patches in the AoI was conducted based on a Matlab script (by Tobias Koch) applying a state–of–the–art image registration technique (Lowe 2004). Since radiometric differences between the different acquisition times and image distortions (e.g. clouds) could remain in the images, feature–based registration methods are preferable over correlation–based registration methods due to their ability to match local feature points instead of entire image areas.

To ensure that actual terrain movements in the AoI do not cause undesired shifts in the registration, the AoI was excluded from the feature point detection step. The remaining feature points were used for estimating a geometric similiarity transformation between the reference and all target images including a statistical outlier removal (RANSAC). This transformation was finally used to accurately register a target image towards the reference image.

Regarding the registration quality in the test site, a satisfying amount of feature matches of at least 500 after outlier removal could be found for all reference (master) and target (slave) image pairs and for all investigated sensors. The mean distance of transformed inlier feature points of the target image to their corresponding feature matches in the reference image ranged between 0.6 and 0.8 pixels, confirming the high registration accuracy (see OSM Fig. 14).

- L288/289: Usually matching between consecutive images is not achieved by matching "common pixels", but by maximizing the correlation between pixel-value distributions of patches of pixels (i.e. your windows of different sizes in Tab. 6).

  Yes, you are correct it estimates first the pixelwise displacement between two patches based on correlation peaks and second, the final correlation is performed to retrieve the subpixel displacement.

  We added this information and reordered the processing steps according to the COSI–Corr manual (Ayoub et al. 2009).

  There are two correlators; in the frequency domain based on FFT algorithm (Fast Fourier Transformation) and a statistical one. Applying the more accurate frequential correlator engine, recommended for optical images, different parameter combinations of window sizes, direction step sizes and robustness iterations were tested.

  Parameter settings include the initial window size for the estimation of the pixelwise displacement between the images and the final window size for subpixel displacement computation in x, y; a direction step in x, y between the sliding windows; and several robustness iterations (**Fehler! Verweisquelle konnte nicht gefunden werden.**).

  [...]

  The results of each correlation computation returns a signal–to–noise ratio map (SNR) and displacement fields in east–west and north–south directions. These results were exported from ENVI classic as GTiff, and the total displacement was then calculated with QGIS.

- L304/305: What is the uncertainty of these east-west and north-south displacement estimates? Did you check whether the bearing of the displacement matches the general slope of the Sattelkar?

  In the OSM we are providing the results of the correlation computations for our published results (east–west and north–south displacement fields as well as signal–to–noise maps). The results are consistent. We further provide total displacement results of other parameter combinations.

  Yes, we checked the overall orientation of the correlation based on computed directional vectors (with SAGA GIS software). We provide these vectors in the OSM, too (OSM Fig. 3 (b) and (h)).

- L307/308 and L440/442: This seems a bit arbitrary. How did you determine a cutoff–value of 4m displacement? How did you distinguish outliers from non-outliers? What is the confidence of your estimates?
  We determined the cutoff–value employing several criteria. First based on field experience we know the landslide extent and displayed the results in combination with the demarcation displayed as 'Active area' in Figs. 2, 3, 5 and 7. Then we checked the value distribution in the histograms for both the calculated total displacement as well as the signal–to–noise maps. These maps were further used to visually compare the total displacement results. This allowed us to identify outliers and unlikely displacement. Based on the histograms and the acquired experience for the results, the thresholds were tested and set for transparency and to display values. Please see the OSM (Fig. 13).

- L308/309: This contradicts the descriptions of Fig. 5a, where you point out that "ambiguous, small-scale patterns with highly variable displacement rates" (L332/333) dominate the western part of the mass movement.
  Here we would like to differentiate between inconsistencies which we understand as artefacts and noise due to snow, vegetation, clouds, cloud shadows and terrain shadows. De–correlation with its salt–and–pepper appearance due to velocities exceeding the correlation capability of DIC have a different origin and reason.
  However in the results, section 5, we described the appearance of these ambiguous signals, while in the discussion section they are explained.

- L311/312: I am not convinced that manually tracking boulders in the same images that were used for image correlation can verify the results of this correlation. You can use these data to check if manual and automated tracking give consistent results. Comparing manually tracked boulders from UAS imagery could however be used to compare against the displacement estimates from satellite imagery.
  We are certain that the direct measurements of travelling distances from blocks of 10 m size for consecutive orthoimages, which were also employed for the DIC method, are a valid method to underpin the total displacement results by the DIC.
  Comparing these tracks with satellite imagery might be useful keeping in mind that the difference between UAS orthoimages of 0.16 m and PlanetScope satellite images of 3 m spatial resolutions is substantial and sensor type, image processing etc. can introduce further inaccuracies.

- L320: As you present total displacement for different time intervals here, not rates in distance per unit time, I would suggest changing the title here. Same is true for L326, L346 and L361.
  Yes, thank you for pointing this out. We changed the section title (see below) and in the text accordingly (L326, L346, L350, L354, L357 and L361).
  Section Title: 5.1. Total displacements

- L335/336 and L366: Did you check the direction of displacement for the areas of smallscale patterns of ambiguous signals? I would suspect that these are very heterogenous here as well. It would also be worth looking into the quality information (correlation coefficients) for these regions.

Yes, this is a good point. Indeed, we checked the direction based on displacement vectors as well as signal–to–noise maps. They both give the same indication of heterogeneous and ambiguous signals with no correlation for exactly the same areas with ambiguous signals in the total displacement calculation. Please see our OSM (OSM Fig. 3).

- L397: For a comparison (and also for a better readability) you could convert your total displacement to average rates of m yr-1 or cm d-1.
  Yes, converting them into averaged rates is a good suggestion for the discussion section, see below. If you recommend this conversion for the results section 5 too, then the section title should be kept "Displacement rates" as before (see your previous comment for L320). For the (old) L417, 418 and 420, the values were added with yearly rates in brackets:
  trajectories up to 4 m ($34.8$ m yr$^{-1}$) (d); a 16 m ($139$ m yr$^{-1}$) trajectory (a); approximately 10 m ($86.9$ m yr$^{-1}$)

- L398/399 and L402/404: Given the large differences in total displacement between sensors and resolutions used for image cross-correlation, I do not think that you can make this claim. Please use an appropriate measure to quantify the agreement between manual boulder tracking and the three different approaches used for digital image correlation.
  These lines refer to the results of the total displacement derived from UAS orthophotos. Regarding L398/399 the parameters were tested and selected independent of others' recommendations, but we arrived at the similar conclusions. With regard to L402/404 we believe that the travel distance measurements of field mapped boulders based on the same data (UAS orthophotos) are comparable to DIC derived total displacements.

- L419/422: This might be the case, though you tested larger patch sizes (Tab. 6) that should have given you consistent results for this region then.
  In the OSM we provide results of our parameter tests for larger final window sizes (see OSM Fig. 5 and 7).

- L433/434: This should be backed by a statistical measure. From a close look to Fig. 5, I rather get the impression that the only patches you can make this statement for is location a in Fig. 5 (b) and (d) and location c in Fig. 5 (a) and (c), but to a lesser extent.
  Thank you for pointing this out. In our opinion the first time interval with slightly more than one year of accumulated displacement, the frontal area and core body of the landslide are reflected in both DIC results of UAS and PlanetScope (locations (*a*) and (*c*), as well as (*d*) and slightly (*e*) and (*f*) in Fig. 5 a) for UAS and c) for PlanetScope I). In contrast to the second interval of 42 days, it seems that there is not enough accumulated displacement to be captured by PlanetScope DIC, as the middle to rear landslide body are only reflected in the UAS DIC result (locations (*b*)–(*d*) Fig. 5 c) and remain free of signal for these locations in Fig. 5 d) for PlanetScope.

- L445/447: The size of the snow patches does not play an important role. The presence of snow in one image hampers correlation between images and leads to false patchmatching results.
  Yes, we absolutely agree and this is also described by Leprince et al. (2007; 2008), noting that variations, thus the difference in snow cover, limit the technology. In addition, they say that in images with high gains, the areas of snow coverage are saturated too, and as a result, do not allow for any correlation (Scherler et al. 2008).

Regarding the displacement for (j) as identified in both sensor combinations (see Fig. 5), there is a patch of snow (1–2 m height, length ~ 25 m, see OSM Fig. 10) in the UAS and PlanetScope images on 24.7.2019 while for the images on 13.7.2018/19.7.2018 (UAS/PlanetScope) and 4.9.2019 (UAS and PlanetScope) there is no snow (see OSM Fig. 2 and 11). Thus, in this case, the existence of snow in one image but not in the other explains this false correlation and indication of displacement.

Minor snow fields as visible in the images from 24.07.2019 for both, UAS and PlanetScope, likely explain the big cluster of incorrect displacement southeast of the lobe (*j*); nonetheless, in the satellite image they are smaller than the resulting DIC displacement.

- L457/462: To be frank, I do not see much similarity between Fig. 5 (c) and (e) nor (d) and (f). I would be very cautious in interpreting these results as is. This is especially true for the resampled UAS results.
  Thank you for pointing this out. Yes, we agree in some part. Our purpose was to compare our high accuracy UAS orthophotos to PlanetScope satellite images, in order to estimate the goodness of fit and limitations of the latter.
  We are aware that this downsampling factor is large, and therefore the resulting displacement rates and inherent velocities have to be viewed with reservations.

  However, in terms of noise outside our defined active landslide area and the overall detection to the landslide boundary as delineated based on the 0.16 m UAS data: for the first, the noise is low to moderate, and there is generally a good fit for the 3 m downsampled UAS data similar to DIC results of UAS at 0.16 m, respectively. In contrast, DIC results of PlanetScope neither show likewise noise–free areas outside the active landslide regions nor do they reach the same extent total displacement extent as the downsampled UAS data.

- L463/464: As the GCPs for referencing the UAS data are probably located close to the landslide, it is not surprising, but neither disturbing, that false displacement clusters appear outside the area of interest.
  Please see our map of GCP distribution as well as images thereof in the OSM (OSM Fig. 1). Some GCPs are close to the landslide area, but installed on stable bedrock and to best of our knowledge, they are not moving and thus provide continuous usability and comparability.

  False displacement is indicated for a cluster outside of the boundary to the image border in the east for UAS interval I (Fig. 5e) and in the north western area (*h*, *i*) for interval II (Fig. 5f) contributing to changes in shading and illumination.

- L468/470: Again, I would not trust the displacement estimates of the resampled UAS data. While it is true that your manual boulder tracking identified 2 boulders with displacement of 10 or more meters, the remaining 34 boulders show something different.
  Yes, you are right that not all of the 34 boulders are exactly reflecting the DIC total displacement result. However there are more than two which are in the same range of displacement, and others are very close to it, keeping in mind that there are some uncertainties and limitations when it comes to the threshold of identification of small ground motions in the DIC method. Please see here the section 6.1, discussion. We are happy to revise this further.

- L471/476: While it might be true that the results obtained from image correlation of resampled 3m UAS data are better (internally) correlated and show a more homogeneous deformation pattern, this does not mean that the result is correct. As I outlined above, I have serious doubts regarding the interpretability of this data, as there is no agreement with the manually tracked boulder velocities (except 2 boulders). Also, from a geomorphic perspective, I am not sure how you would explain a velocity pattern where high velocities dominate throughout the entire landslide, but are speckled with lower to zero movement within (Fig. 5 e and f).
  We agree with the 3 m resolution to some extent. Please see comment above for L457/462 the comparability of manual block tracking to UAS DIC result.
  The 'speckled' pattern, is due to decorrelation resulting from velocities too high to be captured with the DIC method; this combined with an observation period of 42 days delay (Delacourt et al. 2007; Travelletti et al. 2012) may be exceeding the accumulated displacement to be captured by the method, which could contribute to this pattern and explain the resulting limitation to some extent. In addition, we know that the surface changes significantly in the frontal part and these strong alterations also limit the DIC method (Lewis 2001; Travelletti et al. 2012). For more please see section 6.1, discussion, too. If this is not clear enough in the discussion, we would be happy to further revise this.

- L485/488: Did you evaluate the proportion of false-positive displacements to truepositive displacements and if so, how did you do this and can you please include this data? Based on the image correlation results shown here, you can make this statement, but I would be cautious to make a general claim on the usability of the data.
  We approached our results by testing of different parameter settings and combinations based on visual comparison as is common in the field (Bontemps et al. 2018). The PlanetScope DIC results presented here are the most suitable master–slave image combinations. We could provide the other intervals of DIC results which are not meaningful for comparison if wished.

- L552/554 / Table 7 / Figure 9: I do like the idea behind this, where the authors show that their proposed workflow would enable a timely warning in the case of historic landslides. However, in the case of Vajont, I think you should include a critical factor. While it is theoretically true that a "forecasting window" would allow for your workflow to be completed well before the failure, the slow deformation of Vajont (35 mm d-1) in the 30 days will be well below the level of detection of your image correlation analysis, if you collect an image directly after the onset of "significant acceleration". In order to be detectable, movement must have accumulated a critical distance before data collection of your workflow can set in (30 days = 1.05 m total displacement) – a factor that in my view would be important to include here.
  Thank you for mentioning this, you are absolutely right. We added the following sentence below to emphasise this critical detection capability limit of the DIC method.
  We assume that approximately 30 days before failure Vajont would have displayed a signal exceeding the noise at modern standards and would have become predictable.
  For Vajont, the 1/velocity plot by Petley and Petley (2006) (based on data from Semenza and Ghirotti (2000)) shows an increase in movement at about day 60 along with a transition from a linear to an asymptotic trend at

approximately day 30, defined as a transition from ductile to brittle. Therefore, we assumed 30 days of forecasting window for twarning and tlead until the impact of the hazardous event on 09.10.1963. However, it has to be kept in mind that velocities of about 35 mm d$^{-1}$ are still low and at the minimum of the displacement recognition capability for the digital image correlation method.

**Technical corrections:**

- L1: Landslide
  Here we are referring to landslides in general, not to a specific landslide.
- L103/105: Check grammar
  We did not add a comma as the text is in BE; in AE, however, a comma could be added (In this investigation,…). We added quotation marks to improve readability.
- L185: Is this really the source the authors need to cite for the location map?
  Thank you, we modified in response to comment by RC1 (J. Blöthe) by changing Vienna to Wien. Otherwise this is according to the publishing company and the copyright statement from the online map.
  **Figure 1** (**a**) Overview map Austria (Österreichischer Bundesverlag Schulbuch GmbH & Co. KG and

  Freytag–Berndt & Artaria KG, Wien).

- L229: beginning of April
  Thank you, we inserted missing word.
  span from the beginning of April to the end of October in 2019

- Table 3: Here you use a different date format than in the text
  Thank you, we corrected the format. In addition to that we also reformatted the dates in Table 6 accordingly.
- L257: UgCS-Software?
  Further information on the flightplanning Software UgCS can be found here:
  https://www.ugcs.com/photogrammetry-tool-for-land-surveying
- Table 4: Unit for GSD missing
  Thank you, we added the GSD unit.
- L273/274: Add this information to Table 5 and delete here
  This is a good suggestion and we followed it.
- L299/300: I guess this is only relevant if you explicitly mention the image–processing times.
  Thank you, however we think this is relevant as the duration of image processing and DIC calculation are an important part of our temporal concept in the results section 5.3. and discussion section 6.3.
- L398: can be compared
  We think that the repetition of 'compared' is not necessary; it follows from the logic of the sentence.
- L409/410: resulting from significant morphological changes?
  Thank you for pointing this out. After a detailed verification of volumetric calculations, we can confirm changes of about 1 m. Please see our calculations and visualisations in the OSM.
  In Fig. 5a, the large southern patch (*g*) shows clear displacement values for the rear part and decorrelation for

  the front region resulting from morphological changes within the image pair of interval I.

- L443: bracket missing?
  Yes you are right, thank you.

- L 460: check figure reference
  Thank you for pointing on this auto–correction mistake.

**References**

Ayoub, Francois; Leprince, Sébastien; Keene, Lionel (2009): User's Guide to COSI-CORR Co-registration of Optically Sensed Images and Correlation. California Institute of Technology. Pasadena, CA 91125, USA.

Bontemps, Noélie; Lacroix, Pascal; Doin, Marie-Pierre (2018): Inversion of deformation fields time-series from optical images, and application to the long term kinematics of slow-moving landslides in Peru. In: *Remote Sensing of Environment* 210, S. 144–158. DOI: 10.1016/j.rse.2018.02.023.

Delacourt, Christophe; Allemand, Pascal; Berthier, Etienne; Raucoules, Daniel; Casson, Bérangère; Grandjean, Philippe et al. (2007): Remote-sensing techniques for analysing landslide kinematics: a review. In: *Bulletin de la Societe Geologique de France* 178 (2), S. 89–100. DOI: 10.2113/gssgfbull.178.2.89.

Leprince, Sébastien; Barbot, Sylvain; Ayoub, Franois; Avouac, Jean-Philippe (2007): Automatic and Precise Orthorectification, Coregistration, and Subpixel Correlation of Satellite Images, Application to Ground Deformation Measurements. In: *IEEE Trans. Geosci. Remote Sensing* 45 (6), S. 1529–1558. DOI: 10.1109/TGRS.2006.888937.

Leprince, Sébastien; Berthier, Etienne; Ayoub, Francois; Delacourt, Christophe; Avouac, Jean-Philippe (2008): Monitoring Earth Surface Dynamics With Optical Imagery. In: *Eos* 89. DOI: 10.1029/2008EO010001.

Lewis, J. P. (2001): Fast Normalized Cross-Correlation. In: *Ind. Light Magic* 10.

Lowe, G. (2004): SIFT-The scale invariant feature transform. In: *Int. J.* 2, S. 91–110.

Scherler, D.; Leprince, Sébastien; Strecker, M. (2008): Glacier-surface velocities in alpine terrain from optical satellite imagery—Accuracy improvement and quality assessment. In: *Remote Sensing of Environment* 112 (10), S. 3806–3819. DOI: 10.1016/j.rse.2008.05.018.

Stumpf, Andre; Michéa, David; Malet, Jean-Philippe (2018): Improved Co-Registration of Sentinel-2 and Landsat-8 Imagery for Earth Surface Motion Measurements. In: *Remote Sensing* 10, S. 160. DOI: 10.3390/rs10020160.

Travelletti, Julien; Delacourt, C.; Allemand, P.; Malet, J.-P; Schmittbuhl, Jean; Toussaint, R.; Bastard, M. (2012): Correlation of multi-temporal ground-based optical images for landslide monitoring: Application, potential and limitations. In: *Journal of Photogrammetry and Remote Sensing* 70, S. 39–55.

---

## Author Comment (AC4)

**Letter of response to comment on nhess-2021-18**

Dear Sigrid Roessner,

We thank you for your valuable comments on our manuscript and appreciate the time and the efforts you have invested. Your feedback has helped us to see and clarify ambiguous areas to further improve our work.

Based on your suggestions we have restructured the entire manuscript, especially introduction, study site description, discussion and conclusion. In addition, we have specified many conceptual and methodological concerns according to your more specific remarks. We have also rephrased several ambiguous paragraphs.

Please find below the following colour coding for the review and your comments in black; our responses to the review are in blue and the changes made to the manuscript are in green (following RC2), orange (following RC1) and in blue by the authors. Reference to line numbers are based on the original preprint.

**General comments**

The paper represents an interesting contribution to process oriented remote sensing based monitoring of complex landslides with the aim of making a conceptual contribution to early warning. The paper is well written in language and structure and the figures are of good quality. Despite the overall good scientific relevance and presentation quality, in the current form the paper lacks a coherent scientific goal justifying the used approach. This problem already becomes apparent in L40 where the authors state that the study presents a new concept to systematically evaluate remote sensing techniques to optimize lead time for landslide early warning'. Although the presented work is very interesting, it does not fit the stated goal for the following reasons:

- Concept of lead time and need for best possible reduction is not new.
  While we agree that the concept itself may not be knew, we find that using multispectral remote sensing products to assess and increase lead time to ensure the timely prediction of landslide early warning systems represents an important research gap that so far has rarely been addressed. We evaluate the capabilities of remote sensing to identify hot–spots and detect process behaviour changes based on the local conditions. Thus, the landslide process is the precondition. We want to estimate, based on the assumption that the particular sensor is able to deliver the necessary information, the time demand of each sensor for time to warning.
  We have now replaced the phrase optimising lead time with a more precise description of what we have done. Please see revision of the conclusion further below.

L10–11: We introduce a novel conceptual approach  to structure and quantitatively assess lead time  for LEWS.

[…]

L39–41: This study presents a new concept to systematically evaluate remote sensing techniques to  estimate and increase lead time for landslide early warnings in these catchments. We do not start from the perspective of available data; instead, we define necessary time constraints to successfully employ remote–sensing data  to   ing early warnings.

[…]

L34: Lead time as defined in the context of LEWS is the interval between the issue of a warning (i.e. dissemination) and the forecasted landslide onset (Pecoraro et al. 2019) and thus crucially depends on time requirements in phases

(1)–(3). The success of an EWS therefore requires measurable pre–failure motion (or slow slope displacement) to allow for sufficient lead time for decisions on reactions and counter measures (Grasso, 2014; Hungr et al., 2014).

- Remote sensing techniques themselves are not the bottleneck for shortening the lead time.
  The goal of our concept is not to refine remote sensing as a technique itself but to provide a tool for choosing the appropriate sensors based on time required for the time to warning phase. We thereby increase lead time.
  We do not agree with your objection to the word "bottleneck" especially given your comment below which says "In remote sensing based approaches lead time mostly depends on the available imaging constellation and data distribution to the end user."

L39–61: This study presents a new concept to systematically evaluate remote sensing techniques to  estimate and increase lead time for landslide early warnings in these catchments. We do not start from the perspective of available data; instead, we define necessary time constraints to successfully employ remote–sensing data  to provid early warnings. This approach reduces  to a small number the  suitable remote sensing products  with high temporal and spatial resolution. With these constraints, we investigated the application of data from satellites and  (UAS to allow the assessment of the data, after a spaceborne area–wide but low–resolution acquisition, into a downscaled detailed image recording. In so doing, we analysed the capability of these different passive remote sensing systems focusing on spatiotemporal capabilities for ground motion detection and landslide evolution to provide early warnings.

[…]

L94–102: In recent years, data provision for users has increased and today data hubs provide easy accessibility to rapid, pre–processed imagery.  Nonetheless, technological advances can be misleading as they promise high spatiotemporal data availability, which frequently does not reflect reality (Sudmanns et al., 2019). One key problem is the realistic net temporal data resolution which is often significantly reduced due to technical issues, such as image errors and non–existent data (i.e. data availability, completeness, reliability). Other problems include data quality and accuracy in terms of geometric, radiometric and spectral factors (Batini et al., 2017; Barsi et al., 2018). Knowledge of the most useful remote sensing data options is vital for complex, time–critical analyses such as ground motion monitoring and landslide early warning. Timely information extraction and interpretation are critical for landslide early warnings yet few studies have so far explicitly focused on time criticality and the influence of the net temporal resolution of remote sensing data.

- In remote sensing based approaches lead time mostly depends on the available imaging constellation and data distribution to the end user and in case of optical data on the atmospheric conditions (clouds). Both factors are only to a very limited extent in control of the authors - only in case of the UAV data acquisitions.
  Thank you for your comment. We agree that the limitation of meteorological conditions including effects such as cloud shadow and snow are important constraints as we described in L45–55 and L158. We took this into consideration when estimating

the number of available PlanetScope images (Sect. 4.2.) and discussed atmospheric affected images with regard to displacement derivation results in L477–481.

You are right that for UAS campaigns, most of the control is on the user side and only to a very limited part for other satellites. Today, some data providers promise new images daily, sometime even more frequently (e.g. PlanetScope).

But this is the point we want to highlight with our study. In a real world situation, we wish to determine which satellites can provide useful timely information in terms of an effective repetition rate and real availability in the data hub (provider). In addition, the natural conditions such as atmospheric and site specific constraints can reduce the net image number. For this reason, we assess the capabilities of optical remote sensors in a spatiotemporal context for given circumstances to detect hot spots and identify possible changes in slope processes.

L52–55: Previously, high spatial resolution satellite data was obtained at the expense of a reduction in the revisit rates (Aubrecht et al., 2017). Consequently, the return period between two images increased, limiting ground displacement assessment and the range of observable motion rates. The number of useful images was further reduced due to natural factors such as snow cover, cloud cover and cloud shadows.

[…]

L86–91: In general, sensor choice depends on the landslide motion rate with radar at the lower and optical instruments at the upper motion range (Crosetto et al., 2016; Moretto et al., 2017; Lacroix et al., 2019). However,  flexible, cost–effective alternative to spaceborne optical data are airborne optical images taken by UASs . Freely selectable flight routes and acquisition dates  enable avoiding shadows from clouds and topographic obstacles as well as  unfavourable weather conditions and summer time snow cover, all of which frequently impair satellite images (Giordan et al., 2018; Lucieer et al., 2014).

L96–102: […] technological advances can be misleading as they promise high spatiotemporal data availability, which frequently does not reflect reality (Sudmanns et al., 2019). One key problem is the realistic net temporal data resolution which is often significantly reduced due to technical issues, such as image errors and non–existent data (i.e. data availability, completeness, reliability). Other problems include data quality and accuracy in terms of geometric, radiometric and spectral factors (Batini et al., 2017; Barsi et al., 2018). Knowledge of the most useful remote sensing data options is vital for complex, time–critical analyses such as ground motion monitoring and landslide early warning. Timely information extraction and interpretation are critical for landslide early warnings yet few studies have so far explicitly focused on time criticality and the influence of the net temporal resolution of remote sensing data.

- The used data sources (planet and UAV) do not allow optimization of lead time in the context of early warning because of the scarcity of their availability which is reflected in the small number of only three multitemporal data takes between July and September analyzed in this study (Table 3)

  Thank you. With regard to this comment we assume this needs further clarification. First, we have changed the entire phrase on "optimising lead time" to be more precise in the description of our approach (see previous comment). Regarding the data takes, yes, we do have three UAS acquisitions but over the course of more than one year (7/2018–9/2019). For the purpose of this comparison we selected PlanetScope data at a similar time to UAS acquisitions, whereby one Planet image (02.07.2018, see Table 5) showed low quality results why the time interval was excluded (see caption Fig. 4). In both UAS and PlanetScope DIC results we can see the general distinctive hot–spot

identification as well as changes in motion behaviour indicating an acceleration for the time intervals I and II. Second, we can obtain a higher frequency of UAS acquisitions if necessary. We have revised our conclusion to be more concise in our work with regard to both, the term optimisation as well as the total number of data takes.

L567–569: This paper presents an innovative concept to compare the lead time for landslide early warnings,  of two optical remote sensing systems. We tested this temporal concept by applying UAS and PlanetScope images of temporal proximity as these are currently the sensors with the best spatiotemporal resolution.

[…]

L573–580: Our findings derived from DIC for this steep high–alpine case study show that high resolution UAS data (0.16 m) can be employed to identify and demarcate the main landslide process and reveal its heterogeneous motion behaviour as confirmed by single block tracking. Thus, validated total displacement ranges from 1–4 m and up to 14 m for 42 days. PlanetScope Ortho Scenes (3 m) can detect the displacement of the landslide central core, however, cannot accurately  represent its extent and internal behaviour. The signal–to–noise ratio, including multiple false–positive displacements, complicates the detection of hotspots at least in this very steep and heterogeneous alpine terrain.

Coarse temporal data resolution, such as in the case study investigated here, represents an important restriction to the use of optical remote sensing data for landslide early warning applications. Acceleration (and the resulting failure) over short periods of time will likely go unnoticed due to large data acquisition intervals. However, for prolonged acceleration periods, such as observed at the Sattelkar slide and many other relevant hazard sites, the chosen data sources have been demonstrated to represent a formidable early warning approach capable of contributing to an improved risk analysis and evaluation in steep high–alpine regions.

[…]

L589–594: For continuous monitoring and early warning, the warning time window could be shortened by on–site drone ports with autonomous acquisition flights and automatic processing. Our systematic evaluation of the sensor  capability can be applied  to other optical remote sensing sensors, and the same is true for our conceptual approach  which extend the lead time. Future studies should focus on the applicability of complementary optical data to confirm the detection of landslide displacement and adjust UAS output resolution as this significantly increases the validity of DIC internal ground motion behaviour.

- The missing sound conceptual approach is also reflected in the introduction in form of a lengthy summary of in principle available remote sensing methods and data showing no clear line of arguments (L20-100). Moreover, the new conceptual approach presented in Fig. 1 is very general and not specific to landslide and does not qualify as a novelty in the current form.

  1. Introduction
  We revised the abstract and the introduction , to be more precise with regard to our goal and implementation. In so doing we more clearly defined our approach to lead time and early warning systems for landslides. Further we did our best to improve the line of arguments and to show the historic limitations of optical remote sensing for LEWS up to the recent developments when it comes to options such as high spatiotemporal products and their usage for monitoring, early warning and time-series displacement analyses.

**2. The conceptual approach**

We decided to keep this concept general, to employ it for other remote sensing techniques and maybe even other kind of instrumentation as well as different use cases of other time challenging issues. We revised and added some sentences to emphasise our approach/idea. Even after intense research we did not find good conceptual approaches challenging remote sensing in the direct context of landslide early warning systems. We therefore consider our approach novel. This concept forms the basis to employ this for the setup of 'a real early warning system'.

[revised manuscript text omitted]

- L140: General applicability to optical data: This subheading does not fit the content of this section comprising a compilation of rather basic and general steps of remote sensing data processing.
  Thank you for your comment. We agree that it describes general steps of the data processing chain; however, these steps are applied within each phase of the 'time to warning' of our proposed concept. Otherwise the steps would not be explained and thus the basis for the concept would be lacking. We have revised the subheading to "Practical implementation of multispectral data in the concept" which more accurately describes the content of this section.

  **2.2. Practical implementation of multispectral data in the concept**

- The study site (starting at L175) represents a very complex landslide case leading to rather erratic mass movements in form of debris flows initiated by changing slope water conditions related to increased atmospheric precipitation. This situation is another obstacle for an early warning approach which is solely based on optical remote sensing data and thus making it impossible to make full use of the in principle daily temporal resolution of the planet data. Taking into account these natural conditions and the constraints introduced by the used imaging constellations, leaves no room for true optimization of lead time in the sense as stated in the overall scientific goal of this paper.
  We agree with your assessment and have replaced the term "optimisation" with a description that hopefully is more accurate in the entire manuscript. The chosen Sattelkar slide is one of the most relevant high-alpine geohazards in Austria and thus represents a compelling study site for natural hazard studies. While we agree that its complexity represents an obstacle, we nonetheless believe that the Sattelkar slide is well-suited for an investigation based on optical remote sensing because (i) we were clearly able to detect significant displacement and (ii) we were able to identify patches of increasing motion. In any case an increase in frequency of UAS flights is possible.

L39–41: This study presents a new concept to systematically evaluate remote sensing techniques to  estimate and increase lead time for landslide early warnings in these catchments. We do not start from the perspective of available data; instead, we define necessary time constraints to successfully employ remote–sensing data  to provid early warnings.

- Any sensible early warning approach for slope movements requires a continuous and reliable high temporal resolution input of observation data related to parameters which are relevant for triggering the potential mass movements. Such information are mostly provided by ground based measurements. In this context, it is surprising that no relevant ground based monitoring information seem to be available to this study despite the longterm history of scientific work at this study site. The mentioned temperature loggers need to be explained in their function for early warning. The GPS measurements seem to only support the remote sensing based analysis. The described setting does not seem to be suitable for identification of precursory signs of ‚slope preparation' related to the triggering of potential mass movements at this site in a way which would be required in the context of early warning.
  Thank you for your feedback. We understand your arguments, yet we are not trying to

create an all-encompassing landslide early warning study that includes all state-of-the-art methods. We have chosen the Sattelkar due to its scientific and societal relevance and its high-alpine location with very limited vegetation. This site was not selected to evaluate a wide range of remote sensing applications. Our goal was to determine if and how our conceptual approach is applicable to this highly complex study site. Due to its topographical characteristics no ground based technique can be implemented. Therefore, only air- and spaceborne sensors can be employed which we believe is the case for numerous potentially hazardous slides/creeps in mountain ranges worldwide. However, we have considered installing a camera on the opposite slope but currently the distance is a problem (3.5 km, selection of camera).

We agree that the temperature data mentioned in the manuscript is not absolutely necessary to understand our conceptual approach. We still think that the (brief) inclusion of the temperature data makes sense as it suggests local permafrost presence/degradation which may be one of the main drivers of the Sattelkar slide. To clarify the role of the temperature data we amended the relevant sections in the study site section.

L175 et seq. […] massive volumes of glacial and periglacial debris as well as rockfall deposits (Fig. 2b, c).

[…] allowing visual block tracking and delimiting the active process area. High displacement was measured between 2012 and 2015 with up to 30 m a$^{-1}$.

[…]

L200 et seq.: In the Sattelkar cirque, several monitoring components are installed to provide ongoing and long–term monitoring. Nine permanent ground control points (GCPs) measured with a dGPS to provide stable and optimal conditions to derive orthophotos from highly accurate UAS images (GeoResearch, 2018). A total number of 15 near surface temperature loggers (buried at 0.1 m depth) recorded annual mean temperatures slightly above the freezing point (1–2 °C) in the period 2016 to 2019. Ground thermal conditions at depth react with significant lag times to recent warming and therefore are primarily determined by climatic conditions of the past (Noetzli et al., 2019). Significantly cooler climatic conditions in previous decades and centuries (Auer et al., 2007) thus likely contributed to the formation of (patchy) permafrost at the Sattelkar cirque. Recent empirical–statistical modelling of permafrost distribution in the Hohe Tauern Range confirms possible permafrost presence at the study site (Schrott et al., 2012).

Correct, the dGPS measurements are only used for repeated UAS campaigns and their data derivation. As described earlier, with our technical approach we were able to not only detect hot spots of total displacement but also to see changes in motion and thus certain areas of accelerating behaviour.

- L210: The complete dismissal of radar data is not justifiable in the current form since the authors only take into account InSAR based deformation analysis and neglect that

the technique of pixel offset tracking can be also be applied to the intensity component of radar data. For the mainly rainfall driven processes at the study site, the integration of radar data seems to be mandatory into any sensible remote sensing based early warning approach, since a combination of optical and radar data is required to establish an as continuous as possible time series of remote sensing observations.

Thank you for mentioning radar data. We have described the application of InSAR/DInSAR in the introduction (L86–91) and placed the argument in section "4.1. Optical Imagery".

For this particular site radar data is not practical. Even if foreshortening and layover effects are a minor issue for this site, the main reason to not include this kind of data is the fact that the velocity shows rates exceeding the limits of radar data leading to a loss of coherence.

L78 et seq.: As  landslides tend to accelerate beyond the deformation rate observable with radar systems before failure, we concentrate on optical image analysis (Moretto et al., 2016). One advantage of optical imagery is its temporally dense data (Table 1) compared to open data radar systems with sensor  repeat frequency  every six days and revisit frequency between three days at the equator, about two days over Europe and less than one day at high latitudes (Sentinel–1, ESA). Optical data allows direct visual impressions from the multispectral representation of the acquisition target and the option to employ this data for further complementary and expert analyses. While active radar systems overcome constraints posed by clouds and do not require daylight, data voids can be significant due to layover or shadowing effects in steep mountainous areas (Mazzanti et al., 2012; Plank et al., 2015; Moretto et al., 2016). Moreover, north/south facing slopes are less suitable, thus limit the range of investigation (Darvishi et al., 2018). In general, sensor choice depends on the landslide motion rate with radar at the lower and optical instruments at the upper motion range (Crosetto et al., 2016; Moretto et al., 2017; Lacroix et al., 2019).

- Moreover, taking into account the goal of lead time optimization, I consider it crucial to also include ground-based live-streamed time-lapse imagery in the proposed remote sensing based early warning approach (for an example see the Khan et al. (2021) paper ,Low-Cost Automatic Slope Monitoring Using Vector Tracking Analyses on Live-Streamed Time-Lapse Imagery' published in Remote Sensing).

   Thank you for this idea and forwarding the information on the article of this useful approach for the 'Rest and Be Thankful slope', Scotland, with PIV on time–lapse imagery. For the Sattelkar we conducted preliminary investigations regarding the installation of a camera on the opposite slope. Due to the steep slope the camera would have to be mounted at the same altitude. This means a camera would have to be able to cover a horizontal distance of about 3.5 km. There is a higher chance of mobile network signal which is otherwise unavailable beginning at the entrance of the valley. Nevertheless, the power supply and issues such as rain drops and general pollution on the lense pose problems as Khan et al. (2021) also acknowledge.

The materials and methods section (4.) as well as the result section (5) are sound and well written. Since reviewer 1 has already focused on this part of the paper as well as the accuracy assessment and made detailed suggestions for improving these parts, I only have a few comments left to make on these aspects of the paper.

- L355: The authors state that core areas of the landslide are surrounded by wide fringes with no data. In this context the meaning of the term ,no data' is not clear to me.

Please, explain, what do you mean by 'no data' – either missing results or zero deformation.

Thank you for pointing this out. Here by 'no data' we mean that there is zero deformation and we have revised the text accordingly.

L354 et seq.: No motion was present in a fringe zone along the landslide front (west boundary), similar to results in Fig. 5a and Fig. 5b. In general, the displacement patterns are less smooth than at 0.16 m input resolution. Outside the landslide significant displacements exist at the eastern image border (Fig. 5e) and towards the west (*h*, *i*) (Fig. 5f). In comparison, total displacement  derived from PlanetScope cover in large parts the active area for Ib (Fig. 5c); however, for II only the core area of the landslide shows displacement. In both results the core areas of the landslide are surrounded by wide fringes with zero deformation.

- L370: Fig 6. The obtained deformation results show a very different degree of detail throughout the landslide. For better evaluation of the reasons for these differences the inclusion of an RGB UAV image of the same area would be helpful in order to be able to include surface texture properties in the evaluation of the obtained differences in the deformation patterns.

  Thank you for your good suggestion. We added the corresponding master and slave image below the presented DIC result. The caption has been adjusted accordingly.

[Figure]

**Figure 1 (a)** Displacement derived from UAS data at 0.16 m resolution for interval II (24.07.2019–04.09.2019, 42 d) combined with boulder trajectories (in metres) manually measured in the UAS orthophotos in the same time period. The solid black line represents the boundary of the active landslide based on field mapping. Background: UAS hillshade, 24.07.2019 (0.08 m), orientation -3° from north. UAS orthophotos at 0.16 m resolution for the master **(b)** and slave image **(c)** for the corresponding time interval.

- Conclusions related to the results presented until L370: The presented specific deformation results obtained from the analyzed planet and UAV data, represent a valuable contribution towards an improved area-wide process understanding of so far unprecedented detail for this study site. Conceptually, such investigations mainly contribute to the preparedness phase within the disaster management cycle. Continuation of monitoring of the study site using the described approach would represent a very valuable prerequisite for developing and setting up a true early warning system for this site combining ground based and remote sensing observations. However, the results presented in this paper do not allow optimization of lead times within an early warning approach being stated being as the goal of this paper.
Our approach is not to set up a comprehensive early warning system, which includes all four elements defined by the UNISDR (2006) (see L35–38).
We agree that optimisation of lead time does not accurately represent what we have done in our study. Thus we have revised our manuscript to make it more precise (see

changes to the manuscript here on p. 1, 3–4). Our concept enables us to evaluate lead time based on our proposed structure.

Introduction, L10–11: We introduce a novel conceptual approach  to structure and quantitatively assess lead time  for LEWS.

[…]

L39–41: This study presents a new concept to systematically evaluate remote sensing techniques to  estimate and increase lead time for landslide early warnings in these catchments. We do not start from the perspective of available data; instead, we define necessary time constraints to successfully employ remote–sensing data  to provid early warnings.

[…]

Conclusion, L578 et seq.: Coarse temporal data resolution, such as in the case study investigated here, represents an important restriction to the use of optical remote sensing data for landslide early warning applications. Acceleration (and the resulting failure) over short periods of time will likely go unnoticed due to large data acquisition intervals. However, for prolonged acceleration periods, such as observed at the Sattelkar slide and many other relevant hazard sites, the chosen data sources have been demonstrated to represent a formidable early warning approach capable of contributing to an improved risk analysis and evaluation in steep high–alpine regions.

- L375: 5.3 Time required for collection, processing and evaluation. The presented analysis is rather meaningless, since the scarcity of the available time steps does not allow the detection of critical process stages. Taking into account the big temporal gaps between the data acquisitions, the time needed for handling the planet and UAV imagery is not really relevant for lead time optimization. The obtained times only allow a relative comparison between planet and UAV based data acquisition within the narrow limits of the chosen approach. However, true early warning would require setting up a *semi-automated processing chain* including automated download and screening of available remote sensing data as well as semi-automated subsequent deformation analysis reducing data handling time to a minimum. Under such conditions, primary remote sensing data availability becomes the crucial decisive factor determined by the data distribution procedures of the satellite data providers and the atmospheric conditions in case of optical imagery. In conclusion, it needs to be stated that the used parameter of time to warning is only applicable under the condition of a near real time continuous data stream of input information which is not available within the presented study.
  Thank you for your comment which helps to clarify your understanding of our text. We did not intend to create a 'true early warning' as you described. This was not the goal of our study. The repeated measurements allow the detection of spatial and temporal acceleration patterns and we believe the repeated measurements can be scaled to early warning demands. With regard to your comment on a *semi-automated processing chain* we do not fully agree. Based on our knowledge, even in case of most geotechnical investigations, the data is analysed by experts prior to issuing an early warning (e.g. https://www.bgu.tum.de/landslides/alpsense/projekt/, Leinauer et al. (2020): DOI: 10.1002/geot.202000027).

- L390: In the current form of the paper the points raised in the discussion (6.) are only relevant in the frame of a process-oriented study and not for early warning purposes

since the latter one requires the identification of precursors for critical process stages – tipping points – which are likely to trigger substantial complex mass movements later turning into potentially catastrophic debris flows.

It is our understanding, we can only provide early warnings for processes we understand. The processual understanding is key to anticipating the magnitude, timing, and reach of alpine hazards, thus processual understanding and early warning cannot be separated.

- L490: Estimating time to warning (6.3). This part of the discussion also suffers from the conceptual limitations which have already been pointed out earlier in this review. A comparison of lead times between the different example landslides would only be meaningful in case of continuous high resolution temporal information on deformation allowing the identification of precursory events which is usually only possible using ground based observations. The presented comparison between potential repeat rates of remote sensing data acquisitions and retrospectively derived lead times is too simplistic (Fig. 8), since the main remaining question is, whether the relevant deformation (cracks etc.) can be first, resolved by the used imagery and second, distinguished from other surface disturbances by the used analysis methods.

In this paper, in contrast to remote sensing papers, the time scale required for effective early warnings is given by nature, i.e., the typical acceleration patterns of particular landslides.

With regard to the comparison of historic events, we referred to their natural landslide processes which delimits the possible lead time. Unfortunately, a comparison to these historic examples is limited to a retrospective view. We agree with you regarding the detection of relevant deformations. If the sensors evaluated here could have identified the motion excluded disturbances, then in this temporal concept UAS and PlanetScope would have been able to show an acceleration in a timely fashion.

We want to keep this concept simple to allow the transfer for required processing times from other sensors. The main question is, if the time is sufficient for the whole processing prior to landslide release.

L:148–149 Natural processes and  their developments constantly take place independently, thus dictate the technical approaches and methodologies researchers  can and must apply within a certain time period.

**Overall recommendation:**
The presented results comprise a very interesting process-oriented study evaluating the use of planet and UAV imagery for the derivation of spatiotemporally differentiated deformation information for a rather large and topographically pronounced terrain affected by complex mass wasting processes. I consider these findings well worth being published in this journal. However, the publication of these specific results requires a major conceptual reframing of the work which is targeted at the real potential usability of these results which cannot be early warning because of the reasons already stated in this review.

However, the work presented in this study has the potential to form an important basis for the development of a true early warning concept / approach in the future combining remote sensing and ground based observations targeting at the same parameters allowing a multi-scale assessment of surface deformation related to triggering potential catastrophic mass movements at the study site.

---

## Author Response (AR1)

**Letter of response to comment on nhess-2021-18**

Dear Sigrid Roessner,

We thank you for your valuable comments on our manuscript and appreciate the time and the efforts you have invested. Your feedback has helped us to see and clarify ambiguous areas to further improve our work.

Based on your suggestions we have restructured the entire manuscript, especially introduction, study site description, discussion and conclusion. In addition, we have specified many conceptual and methodological concerns according to your more specific remarks. We have also rephrased several ambiguous paragraphs.

Please find below the following colour coding for the review and your comments in black; our responses to the review are in blue and the changes made to the manuscript are in green (following RC2), orange (following RC1) and in blue by the authors. Reference to line numbers are based on the original preprint.

**General comments**

The paper represents an interesting contribution to process oriented remote sensing based monitoring of complex landslides with the aim of making a conceptual contribution to early warning. The paper is well written in language and structure and the figures are of good quality. Despite the overall good scientific relevance and presentation quality, in the current form the paper lacks a coherent scientific goal justifying the used approach. This problem already becomes apparent in L40 where the authors state that the study presents a new concept to systematically evaluate remote sensing techniques to optimize lead time for landslide early warning'. Although the presented work is very interesting, it does not fit the stated goal for the following reasons:

 Concept of lead time and need for best possible reduction is not new. While we agree that the concept itself may not be knew, we find that using multispectral remote sensing products to assess and increase lead time to ensure the timely prediction of landslide early warning systems represents an important research gap that so far has rarely been addressed. We evaluate the capabilities of remote sensing to identify hot–spots and detect process behaviour changes based on the local conditions. Thus, the landslide process is the precondition. We want to estimate, based on the assumption that the particular sensor is able to deliver the necessary information, the time demand of each sensor for time to warning.
 We have now replaced the phrase optimising lead time with a more precise description of what we have done. Please see revision of the conclusion further below.

L10–11: We introduce a novel conceptual approach for comprehensive to structure and quantitatively assess lead time assessment and optimisation for LEWS.

[...]

L39–41: This study presents a new concept to systematically evaluate remote sensing techniques to optimise estimate and increase lead time for landslide early warnings in these catchments. We do not start from the perspective of available data; instead, we define necessary time constraints to successfully employ remote–sensing data for to provide ing-early warnings.

[...]

L34: Lead time as defined in the context of LEWS is the interval between the issue of a warning (i.e. dissemination) and the forecasted landslide onset (Pecoraro et al. 2019) and thus crucially depends on time requirements in phases

(1)–(3). The success of an EWS therefore requires measurable pre–failure motion (or slow slope displacement) to allow for sufficient lead time for decisions on reactions and counter measures (Grasso, 2014; Hungr et al., 2014).

• Remote sensing techniques themselves are not the bottleneck for shortening the lead time.

The goal of our concept is not to refine remote sensing as a technique itself but to provide a tool for choosing the appropriate sensors based on time required for the time to warning phase. We thereby increase lead time.

We do not agree with your objection to the word "bottleneck" especially given your comment below which says "In remote sensing based approaches lead time mostly depends on the available imaging constellation and data distribution to the end user."

L39–61: This study presents a new concept to systematically evaluate remote sensing techniques to optimise estimate and increase lead time for landslide early warnings in these catchments. We do not start from the perspective of available data; instead, we define necessary time constraints to successfully employ remote–sensing data for-to provideing early warnings. This approach reduces the to a small number the of suitable remote sensing products to a small number with high temporal and spatial resolution. With these constraints, we investigated the application of data from satellites and unmanned aerial systems (UAS) to allow the assessment of the data, after a spaceborne area–wide but low–resolution acquisition, into a downscaled detailed image recording. In so doing, we analysed the capability of these different passive remote sensing systems focusing on spatiotemporal capabilities for ground motion detection and landslide evolution to provide early warnings.

**[...]**

L94–102: In recent years, data provision for users has increased and today data hubs provide easy accessibility to rapid, pre–processed imagery. Knowledge of the most useful remote sensing data options is vital for complex, time–critical analyses such as ground motion monitoring and landslide early warning. Nonetheless, technological advances can be misleading as they promise high spatiotemporal data availability, which frequently does not reflect reality (Sudmanns et al., 2019). One key problem is the realistic net temporal data resolution which is often significantly reduced due to technical issues, such as image errors and non–existent data (i.e. data availability, completeness, reliability). Other problems include data quality and accuracy in terms of geometric, radiometric and spectral factors (Batini et al., 2017; Barsi et al., 2018). Knowledge of the most useful remote sensing data options is vital for complex, time–critical analyses such as ground motion monitoring and landslide early warning. Timely information extraction and interpretation are critical for landslide early warnings yet few studies have so far explicitly focused on time criticality and the influence of the net temporal resolution of remote sensing data.

In remote sensing based approaches lead time mostly depends on the available imaging constellation and data distribution to the end user and in case of optical data on the atmospheric conditions (clouds). Both factors are only to a very limited extent in control of the authors - only in case of the UAV data acquisitions. Thank you for your comment. We agree that the limitation of meteorological conditions including effects such as cloud shadow and snow are important constraints as we described in L45–55 and L158. We took this into consideration when estimating

the number of available PlanetScope images (Sect. 4.2.) and discussed atmospheric affected images with regard to displacement derivation results in L477–481. You are right that for UAS campaigns, most of the control is on the user side and only to a very limited part for other satellites. Today, some data providers promise new images daily, sometime even more frequently (e.g. PlanetScope). But this is the point we want to highlight with our study. In a real world situation, we wish to determine which satellites can provide useful timely information in terms of an effective repetition rate and real availability in the data hub (provider). In addition, the natural conditions such as atmospheric and site specific constraints can reduce the net image number. For this reason, we assess the capabilities of optical remote sensors in a spatiotemporal context for given circumstances to detect hot spots and identify possible changes in slope processes.

L52–55: Previously, high spatial resolution satellite data was obtained at the expense of a reduction in the revisit rates (Aubrecht et al., 2017). Consequently, the return period between two images increased, limiting ground displacement assessment and the range of observable motion rates. The number of useful images was further reduced due to natural factors such as snow cover, cloud cover and cloud shadows.

[...]

L86–91: In general, sensor choice depends on the landslide motion rate with radar at the lower and optical instruments at the upper motion range (Crosetto et al., 2016; Moretto et al., 2017; Lacroix et al., 2019). However, Aa flexible, cost–effective alternative to spaceborne optical data are airborne optical images taken by UASs (unmanned aerial systems). Freely selectable flight routes and acquisition dates prevent–enable avoiding shadows from clouds and topographic obstacles, and as well as allow avoiding unfavourable weather conditions and summer time snow cover, all of which frequently impair satellite images (Giordan et al., 2018; Lucieer et al., 2014).

L96–102: [...] technological advances can be misleading as they promise high spatiotemporal data availability, which frequently does not reflect reality (Sudmanns et al., 2019). One key problem is the realistic net temporal data resolution which is often significantly reduced due to technical issues, such as image errors and non–existent data (i.e. data availability, completeness, reliability). Other problems include data quality and accuracy in terms of geometric, radiometric and spectral factors (Batini et al., 2017; Barsi et al., 2018). Knowledge of the most useful remote sensing data options is vital for complex, time–critical analyses such as ground motion monitoring and landslide early warning. Timely information extraction and interpretation are critical for landslide early warnings yet few studies have so far explicitly focused on time criticality and the influence of the net temporal resolution of remote sensing data.

• The used data sources (planet and UAV) do not allow optimization of lead time in the context of early warning because of the scarcity of their availability which is reflected in the small number of only three multitemporal data takes between July and September analyzed in this study (Table 3)

Thank you. With regard to this comment we assume this needs further clarification. First, we have changed the entire phrase on "optimising lead time" to be more precise in the description of our approach (see previous comment). Regarding the data takes, yes, we do have three UAS acquisitions but over the course of more than one year (7/2018–9/2019). For the purpose of this comparison we selected PlanetScope data at a similar time to UAS acquisitions, whereby one Planet image (02.07.2018, see Table 5) showed low quality results why the time interval was excluded (see caption Fig. 4). In both UAS and PlanetScope DIC results we can see the general distinctive hot–spot

identification as well as changes in motion behaviour indicating an acceleration for the time intervals I and II. Second, we can obtain a higher frequency of UAS acquisitions if necessary. We have revised our conclusion to be more concise in our work with regard to both, the term optimisation as well as the total number of data takes.

L567–569: This paper presents an innovative concept to compare the lead time for landslide early warnings, utilising of two optical remote sensing systems. We tested this temporal concept by applying UAS and PlanetScope images of temporal proximity as these are currently the sensors with the best spatiotemporal resolution.

L573–580: Our findings derived from DIC for this steep high–alpine case study show that high resolution UAS data (0.16 m) can be employed to identify and demarcate the main landslide process and reveal its heterogeneous motion behaviour as confirmed by single block tracking. Thus, validated total displacement ranges from 1–4 m and up to 14 m for 42 days. PlanetScope Ortho Scenes (3 m) can detect the displacement of the landslide central core, however, cannot accurately resolve represent its extent and internal behaviour. The signal–to–noise ratio, including multiple false–positive displacements, complicates the detection of hotspots at least in this very steep and heterogeneous alpine terrain.

Coarse temporal data resolution, such as in the case study investigated here, represents an important restriction to the use of optical remote sensing data for landslide early warning applications. Acceleration (and the resulting failure) over short periods of time will likely go unnoticed due to large data acquisition intervals. However, for prolonged acceleration periods, such as observed at the Sattelkar slide and many other relevant hazard sites, the chosen data sources have been demonstrated to represent a formidable early warning approach capable of contributing to an improved risk analysis and evaluation in steep high–alpine regions.

[...]

L589–594: For continuous monitoring and early warning, the warning time window could be shortened by on–site drone ports with autonomous acquisition flights and automatic processing. Our systematic evaluation of the sensor potency capability can be applied and transferred to other optical remote sensing sensors, and the same is true for our conceptual approach optimising which extendsing the lead time. Future studies should focus on the applicability of complementary optical data to confirm the detection of landslide displacement and adjust UAS output resolution as this significantly increases the validity of DIC internal ground motion behaviour.

• The missing sound conceptual approach is also reflected in the introduction in form of a lengthy summary of in principle available remote sensing methods and data showing no clear line of arguments (L20-100). Moreover, the new conceptual approach presented in Fig. 1 is very general and not specific to landslide and does not qualify as a novelty in the current form.

1. Introduction

We revised the abstract and the introduction , to be more precise with regard to our goal and implementation. In so doing we more clearly defined our approach to lead time and early warning systems for landslides. Further we did our best to improve the line of arguments and to show the historic limitations of optical remote sensing for LEWS up to the recent developments when it comes to options such as high spatiotemporal products and their usage for monitoring, early warning and time-series displacement analyses.

**2. The conceptual approach**

We decided to keep this concept general, to employ it for other remote sensing techniques and maybe even other kind of instrumentation as well as different use cases of other time challenging issues. We revised and added some sentences to emphasise our approach/idea. Even after intense research we did not find good conceptual approaches challenging remote sensing in the direct context of landslide early warning systems. We therefore consider our approach novel. This concept forms the basis to employ this for the setup of 'a real early warning system'.

L21–102: Landslides are a major natural hazard leading to human casualties and socio–economic impacts, mainly by causing infrastructure damage (Dikau et al., 1996; Hilker et al., 2009). They are often triggered by earthquakes, intense short–period or prolonged precipitation, and human activities (Hungr et al., 2014; Froude and Petley, 2018). In a systematic review Gariano and Guzzetti (2016) report in a review study that 80 % of the papers examined papers show causal relationships between landslides and climate change. The ongoing warming of the climate (IPCC, 2014) is likely to decrease slope stability and increase landslide activity (Huggel et al., 2012; Seneviratne et al., 2012), which .This indicates a vital need to improve the ability to detect, monitor and issue early warnings of landslides and thus to reduce and mitigate landslide risk.

Early warning, as defined by the UN International Strategy for Disaster Reduction (UNISDR), refers to a set of capacities for the timely and effective provision of warning information through institutions, such that individuals, communities and organisations exposed to a hazard are able to take action with sufficient time to reduce or avoid risk and prepare an effective response (UNISDR, 2009). According to UNISDR (2006), an effective early warning system consists of four elements: (1) risk knowledge, the systematic data collection and risk assessment; (2) the monitoring and warning service; (3) the dissemination and communication of risk as well as early warnings; and (4) the response capabilities on local and national levels. Incompleteness or failure of one element can lead to a breakdown of the entire system (ibid.). Lead time as defined in the context of LEWS is the interval between the issue of a warning (i.e. dissemination) and the forecasted landslide onset (Pecoraro et al. 2019) and thus crucially depends on time requirements in phases (1)–(3). The success of an EWS therefore requires measurable pre–failure motion (or slow slope displacement) to allow for sufficient lead time for decisions on reactions and counter measures (Grasso, 2014; Hungr et al., 2014).

While remote sensing has been established for early warnings, remote sensing is not yet used for real early warnings of the onset of landslides in steep-alpine terrain (with a few exceptions), where geotechnical instruments are still preferred. Exceptions include terrestrial InSAR (Pesci et al., 2011; Walter et al. 2020) and terrestrial laser scanning with high repetition rates. However, repeated UAS (unmanned aerial systems) and optical satellite images (PlanetScope) with high repetition rates have so far not been applied for landslide early warning in steep-alpine catchments. In this regard, knowledge of sensor capabilities and limitations is essential, as it determines which rates and magnitudes of pre-failure motion can potentially be identified (Desrues et al., 2019). Our proposed framework refers to mass movements in steep–alpine catchments with significant pre–failure motion operating over a-sufficient time periods and thus excludes instantaneous events triggered by processes such as heavy rainfalls or earthquakes.

This study presents a new concept to systematically evaluate remote sensing techniques to optimise estimate and increase lead time for landslide early warnings in these catchments. We do not start from the perspective of available data; instead, we define necessary time constraints to successfully employ remote–sensing data for-to provideing early warnings. This approach reduces the to a small number the of suitable remote sensing products

to a small-with high temporal and spatial resolution. With these constraints, we investigated the application of data from satellites and unmanned aerial systems (UAS) to allow the assessment of the data, after a spaceborne area–wide but low–resolution acquisition, into a downscaled detailed image recording. In so doing, we analysed the capability of these different passive remote sensing systems focusing on spatiotemporal capabilities for ground motion detection and landslide evolution to provide early warnings.

Until Recently, the spatial and temporal resolution of optical satellite imagery has significantly improved requirements for accurate early warning purposes have not been met by optical satellite imagery (Scaioni et al., 2014) and has allowed substantial advances in the definition of displacement rates and acceleration thresholds to approach requirements for early warning purposes. This is essential since spatial and temporal resolution determines whether landslide monitoring is possible with the detection allows defining of displacement rates and the approximation approximate acceleration thresholds, both of which are lacking if information is based solely on post-event studies (Reid et al., 2008; Calvello, 2017). Landslide monitoring offers the potential to significantly advance landslide early warning systems (LEWS) (Chae et al., 2017; Crosta et al., 2017). Previously, high spatial resolution satellite data was obtained at the expense of a reduction in the revisit rates (Aubrecht et al., 2017). Consequently, the return period between two images increased, limiting ground displacement assessment and the range of observable motion rates. The number of useful images was further reduced due to natural factors such as snow cover, cloud cover and cloud shadows. High-resolution remote sensing data was long restricted due to high costs and data volume (Goodchild, 2011; Westoby et al., 2012). Today Ccommercial very high resolution (VHR) optical satellites exist, but tasked acquisitions make them inflexible and very cost intensive, thus limiting research (Butler, 2014; Lucieer et al., 2014). There is a vast spectrum of available remote sensing data with high spatiotemporal resolution (Table 1). Complementary use of different remote sensing sources can significantly improve landslide assessment as demonstrated by Stumpf et al. (2018) and Bontemps et al. (2018), who draw on archive data and utilise different sensor combinations to analyse the evolution of ground motion.

**Table 1** Overview of different optical multispectral remote sensors with their corresponding resolution [m] and revisit rate [days]. The sensors are categorised into commercial and free data policy. 1free quota via Planet Labs Education and Research Program, 2PlanetScope Ortho Scene Product, Level 3B/Ortho Tile Product, Level 3A (Planet Labs, 2020b), 3reached end of life, 3/2020, archive data usable, 45 m Ortho Tile Level 3A (Planet Labs, 2020a), 50.5 m colour pansharpened, 6self–acquired. Source: (ESA, 2020).

[revised manuscript text omitted]

Thank you for your comment. We agree that it describes general steps of the data processing chain; however, these steps are applied within each phase of the 'time to warning' of our proposed concept. Otherwise the steps would not be explained and thus the basis for the concept would be lacking. We have revised the subheading to "Practical implementation of multispectral data in the concept" which more accurately describes the content of this section.

2.2. Practical implementation of multispectral data in the concept General applicability to optical data

• The study site (starting at L175) represents a very complex landslide case leading to rather erratic mass movements in form of debris flows initiated by changing slope water conditions related to increased atmospheric precipitation. This situation is another obstacle for an early warning approach which is solely based on optical remote sensing data and thus making it impossible to make full use of the in principle daily temporal resolution of the planet data. Taking into account these natural conditions and the constraints introduced by the used imaging constellations, leaves no room for true optimization of lead time in the sense as stated in the overall scientific goal of this paper.

We agree with your assessment and have replaced the term "optimisation" with a description that hopefully is more accurate in the entire manuscript. The chosen Sattelkar slide is one of the most relevant high-alpine geohazards in Austria and thus represents a compelling study site for natural hazard studies. While we agree that its complexity represents an obstacle, we nonetheless believe that the Sattelkar slide is well-suited for an investigation based on optical remote sensing because (i) we were clearly able to detect significant displacement and (ii) we were able to identify patches of increasing motion. In any case an increase in frequency of UAS flights is possible.

L39–41: This study presents a new concept to systematically evaluate remote sensing techniques to optimise estimate and increase lead time for landslide early warnings in these catchments. We do not start from the perspective of available data; instead, we define necessary time constraints to successfully employ remote–sensing data for to provide ing-early warnings.

• Any sensible early warning approach for slope movements requires a continuous and reliable high temporal resolution input of observation data related to parameters which are relevant for triggering the potential mass movements. Such information are mostly provided by ground based measurements. In this context, it is surprising that no relevant ground based monitoring information seem to be available to this study despite the longterm history of scientific work at this study site. The mentioned temperature loggers need to be explained in their function for early warning. The GPS measurements seem to only support the remote sensing based analysis. The described setting does not seem to be suitable for identification of precursory signs of ,slope preparation' related to the triggering of potential mass movements at this site in a way which would be required in the context of early warning.

Thank you for your feedback. We understand your arguments, yet we are not trying to

create an all-encompassing landslide early warning study that includes all state-of-theart methods. We have chosen the Sattelkar due to its scientific and societal relevance and its high-alpine location with very limited vegetation. This site was not selected to evaluate a wide range of remote sensing applications. Our goal was to determine if and how our conceptual approach is applicable to this highly complex study site. Due to its topographical characteristics no ground based technique can be implemented. Therefore, only air- and spaceborne sensors can be employed which we believe is the case for numerous potentially hazardous slides/creeps in mountain ranges worldwide. However, we have considered installing a camera on the opposite slope but currently the distance is a problem (3.5 km, selection of camera).

We agree that the temperature data mentioned in the manuscript is not absolutely necessary to understand our conceptual approach. We still think that the (brief) inclusion of the temperature data makes sense as it suggests local permafrost presence/degradation which may be one of the main drivers of the Sattelkar slide. To clarify the role of the temperature data we amended the relevant sections in the study site section.

L175 et seq. [...] massive volumes of glacial and periglacial debris as well as rockfall deposits (Fig. 2b, c).

Near-surface temperature data indicates sporadic permafrost distribution in the upper part of the cirque.

[...] allowing visual block tracking and delimiting the active process area. High displacement was measured between 2012 and 2015 with up to 30 m a-1.

[...]

L200 et seq.: In the Sattelkar cirque, several monitoring components are installed to provide ongoing and longterm monitoring. Nine permanent ground control points (GCPs) measured with a dGPS to provide stable and optimal conditions to derive orthophotos from highly accurate UAS images (GeoResearch, 2018). A total number of 15 near surface temperature loggers (buried at 0.1 m depth) recorded annual mean temperatures slightly above the freezing point (1–2 °C) in the period 2016 to 2019. Ground thermal conditions at depth react with significant lag times to recent warming and therefore are primarily determined by climatic conditions of the past (Noetzli et al., 2019). Significantly cooler climatic conditions in previous decades and centuries (Auer et al., 2007) thus likely contributed to the formation of (patchy) permafrost at the Sattelkar cirque. Recent empirical–statistical modelling of permafrost distribution in the Hohe Tauern Range confirms possible permafrost presence at the study site (Schrott et al., 2012).

These components include 30 near surface temperature logger (NSTL) nine permanent ground control points (GCP) measured with a dGPS to provide stable and optimal conditions for the derivation of orthophotos from highly accurate UAS images (GeoResearch, 2018). Field based mapping and measurements help to delimit the active process area.

Correct, the dGPS measurements are only used for repeated UAS campaigns and their data derivation. As described earlier, with our technical approach we were able to not only detect hot spots of total displacement but also to see changes in motion and thus certain areas of accelerating behaviour.

• L210: The complete dismissal of radar data is not justifiable in the current form since the authors only take into account InSAR based deformation analysis and neglect that

the technique of pixel offset tracking can be also be applied to the intensity component of radar data. For the mainly rainfall driven processes at the study site, the integration of radar data seems to be mandatory into any sensible remote sensing based early warning approach, since a combination of optical and radar data is required to establish an as continuous as possible time series of remote sensing observations. Thank you for mentioning radar data. We have described the application of InSAR/DInSAR in the introduction (L86–91) and placed the argument in section "4.1. Optical Imagery".

For this particular site radar data is not practical. Even if foreshortening and layover effects are a minor issue for this site, the main reason to not include this kind of data is the fact that the velocity shows rates exceeding the limits of radar data leading to a loss of coherence.

L78 et seq.: As forecasted-landslides tend to accelerate beyond the deformation rate observable with radar systems before failure, we concentrate on optical image analysis (Moretto et al., 2016). One advantage of optical imagery is its temporally dense data (Table 1) compared to open data radar systems with sensor visits repeat frequency more than every six days and revisit frequency between three days at the equator, about two days over Europe and less than one day at high latitudes (Sentinel–1, ESA). Optical data allows direct visual impressions from the multispectral representation of the acquisition target and the option to employ this data for further complementary and expert analyses. While active radar systems overcome constraints posed by clouds and do not require daylight, data voids can be significant due to layover or shadowing effects in steep mountainous areas (Mazzanti et al., 2012; Plank et al., 2015; Moretto et al., 2016). Moreover, north/south facing slopes are less suitable, thus limit the range of investigation (Darvishi et al., 2018). In general, sensor choice depends on the landslide motion rate with radar at the lower and optical instruments at the upper motion range (Crosetto et al., 2016; Moretto et al., 2017; Lacroix et al., 2019).

Moreover, taking into account the goal of lead time optimization, I consider it crucial to also include ground-based live-streamed time-lapse imagery in the proposed remote sensing based early warning approach (for an example see the Khan et al. (2021) paper ,Low-Cost Automatic Slope Monitoring Using Vector Tracking Analyses on Live-Streamed Time-Lapse Imagery' published in Remote Sensing).
 Thank you for this idea and forwarding the information on the article of this useful approach for the 'Rest and Be Thankful slope', Scotland, with PIV on time-lapse imagery. For the Sattelkar we conducted preliminary investigations regarding the installation of a camera on the opposite slope. Due to the steep slope the camera would have to be mounted at the same altitude. This means a camera would have to be able to cover a horizontal distance of about 3.5 km. There is a higher chance of mobile network signal which is otherwise unavailable beginning at the entrance of the valley. Nevertheless, the power supply and issues such as rain drops and general pollution on the lense pose problems as Khan et al. (2021) also acknowledge.

The materials and methods section (4.) as well as the result section (5) are sound and well written. Since reviewer 1 has already focused on this part of the paper as well as the accuracy assessment and made detailed suggestions for improving these parts, I only have a few comments left to make on these aspects of the paper.

• L355: The authors state that core areas of the landslide are surrounded by wide fringes with no data. In this context the meaning of the term ,no data' is not clear to me.

Please, explain, what do you mean by ,no data' – either missing results or zero deformation.

Thank you for pointing this out. Here by 'no data' we mean that there is zero deformation and we have revised the text accordingly.

L354 et seq.: No motion was present in a fringe zone along the landslide front (west boundary), similar to results in Fig. 5a and Fig. 5b. In general, the displacement patterns are less smooth than at 0.16 m input resolution. Outside the landslide significant displacements exist at the eastern image border (Fig. 5e) and towards the west (h, i) (Fig. 5f). In comparison, total displacement rates derived from PlanetScope cover in large parts the active area for Ib (Fig. 5c); however, for II only the core area of the landslide shows displacement. In both results the core areas of the landslide are surrounded by wide fringes with zero deformation.

• L370: Fig 6. The obtained deformation results show a very different degree of detail throughout the landslide. For better evaluation of the reasons for these differences the inclusion of an RGB UAV image of the same area would be helpful in order to be able to include surface texture properties in the evaluation of the obtained differences in the deformation patterns.

Thank you for your good suggestion. We added the corresponding master and slave image below the presented DIC result. The caption has been adjusted accordingly.

---

## Author Response (AR2)

**Letter of response to report on nhess-2021-18**

Dear editor Mahdi Motagh,

Thank you for your feedback. We now see clearer and understand the issues raised by reviewer #1. We followed the comments and suggestions carefully and addressed each one thoroughly.

We also thank you for your suggestions. We changed the title and extended the discussion by elaborating on how cloud-processing can contribute to shorten $t_{warning}$ by incorporating tools such as Geohazard GEP. Please see below.

Additionally, please find below the following colour coding for the review and the comments from you and referee#1 (Jan Blöthe) in *black italics*; our responses to the review are in blue and the changes made to the manuscript are in orange (following RC1) and purple (following the Editor). Reference to line numbers are based on the original preprint.

*Comments to the Author by Mahdi Motagh:*
*Your paper was reviewed by two experts in this field. Reviewer 1 still noted a number of important shortcomings in the revised version and called for a 3rd review. I also noticed some issues myself. Although Reviewer 2 was satisfied with the revision, it is evident that the manuscript requires another revision. Please do not send us partially improved manuscript and consider all the comments of the Reviewer 1 that he has made on the annotated pdf and carefully address them in the revised version.*

20

*Also from my side, I think the title needs to be changed. In the discussion and results you carefully addressed all shortcomings that may arise when using optical remote sensing for early warning. Therefore, I would not call it a novel approach.*
*Rather you evaluated the potential that exists in optical remote sensing for early warning. Moreover, please elaborate on the discussion how the t_warning could be shortened using cloud-processing solutions such as Geohazard GEP. Although it is based on Sentinel-2, but the platform could be exploited for big landslides and reduces t_warning significantly.*

**Modifications according to the Editor:**

[L1-2] **Challenging the timely prediction of landslide early warning systems with multispectral remote sensing:**

30 **a  conceptual approach tested in the Sattelkar, Austria**

[L565 ff.] For large and long–preparing slope failures, recent developments such as the ESA's Geohazards Exploitation Platform (GEP), developed and operated by Terradue, support on–demand services such as the Thematic Exploitation Platforms (TEPs) and have the potential to decrease $t_{warning}$: The GEP provides an archive of Copernicus' Sentinel–1 and –2, Pléiades and Spot 6/7 data, and access to cloud computing resources to support large scale geohazard mapping and monitoring (Volat et al. 2017; Foumelis et al. 2019; Lacroix et al.). Therefore, the time critical phases of time to collect and time to process, which in our example are attributed to the larger share of the total time requirement for

$t_{warning}$, could be significantly reduced as the data is directly accessible through high performance cloud computing. What remains is the third phase, time to evaluate, where a relatively short time is required, thus $t_{lead}$ is extended.

40  **Modifications according to referee #1**

Dear Jan Blöthe,

We thank you very much for your time you spent once again reading through our revised manuscript, your response to it and the feedback on our previous letter of response (review no. 1).

Your detailed report clarified miscommunications from the first round and helped to further improve our manuscript. Thank you for your confirmations of our suggestions regarding your comments. We apologise for some misunderstandings (e.g. an overlooked spelling mistake) and are sorry for the difficulties due to the lack of line numbers.

50  *Dear Authors,*

*As this is the second round of reviews, I will limit my comments to your answers on my earlier review in the document nhess-2021-18-AC1-supplement (https://doi.org/10.5194/nhess-2021-18-AC1 ) as well as the additional figures provided in the additional supplementary document nhess-2021-18-AC2-supplement (https://doi.org/10.5194/nhess-2021-18-AC2) .*

*Sadly, there are no line numbers provided, so I decided to attach a commented PDF file of the AC1-supplement. Sorry for the inconvenience, but I hope that this way my comments will be easy to follow and to relate to the specific lines in the text.*

60

*Very generally, I thank the authors for following some of my recommendations that in my view already improved the manuscript at this stage. However, regarding my very general comments, mainly raised under B) "Result of image correlation" in the first round of reviews, I still have major concerns. I outlined these concerns in a number of comments in the attached file but want mention the most critical point here once again.*

*In Fig. 5 the authors present the results of digital image correlation of a landslide in the Sattelkar (Austria) for two different time intervals (interval I/Ib (376/370 days): left panels; interval II (42 days): right panels). First, I am still not convinced that the challenging results for the western part of the*
70  *landslide presented in Fig. 5 a-c, more precisely the random distribution of displacement vectors (a, b) and the patchy nature of vector fields shown in c), are highlighted clearly enough in the manuscript. Leaving these results as is (as the authors have done in the first round of reviews), in my view requires a thorough attribution of the limitations of this data – just labelling these areas as decorrelated is not sufficient.*

*Second, my main concern is the treatment of results shown in Fig. 5 e) and f). Here, the authors use the identical data set as for Fig. 5 a) and b), but resampled to 3 m resolution. Yet the results shown in Fig. 5 a) and b) compared to e) and f) are very different. While the former show rather smoothly distributed displacement of ~2-8 m (a) and ~2 m (b) in the eastern part of the landslide, the latter show displacement*

*predominantly exceeding ~18 m for both time intervals, respectively. Furthermore, the appearance of these displacement fields is very patchy and shows no systematic pattern (Fig. 5 e and f). Yet in L371-378 (revised version of the manuscript) and L479-498 the authors present and discuss the data shown in Fig. 5 e) and f) as if these were adding robust information to the study, without any interpretation of why these results are completely different from Fig. 5 a) and b), nor discussing a potential explanation for this observation.*

*There are a number of points indicating that the data presented in e) and f) is problematic here and that a) and b) are rather plausible. I have listed these concerns, numbered a) to d) and highlighted in the attachment, in the general comments of the first round of reviews and these still stand.*

*Kind regards*

*Jan Blöthe*

**General comments**

Comment: 1
Thank you for adding some more details here.
Thank you for your suggestion. We have added more details.

Comment: 2
This is now a very complete supporting material that in large parts documents the approach that you have taken in your study.
Thank you, we are happy to provide some further analyses and research on the OSM.

Comment 3
I do not think that this is the case. Tracking stable surfaces outside the moving area is very well possible and the figures in the OSM (esp. Fig. 13) indicate that the tracking was done for the entire images. Quantifying spurious displacements in stable regions should therefore be possible. Please also see my comment on this matter below (L285/86).
Yes, you are right. We fear that we have had a misunderstanding in terms of residual mismatch outside the landslide area on stable surfaces. Please see here our profiles for interval I and II in the OSM, Fig. 6. There is zero total displacement in the stable regions outside the active landslide area.

Comment: 1
This is a good apporach, but in my view is not sufficient for the data at hand.
Thank you for your comment. We have extended the OSM and hope that it is now sufficient.

Comment: 2
Please, see my comment below

Comment: 3
Let me again try to illustrate this aspect in a bit more detail. The changes made to the manuscript and foremost the inclusion of additional figures into the OSM helps to shed light on this issue. As the velocity

profiles given in Fig. 6 a) nicely show (please add x- and y-axis labels) is that in the area of decorrelated values, the displacement values in the "salt and pepper" domain of the western part of the landslide, are very heterogeneous. Taking the highest peak as an example, the problem of this data becomes apparent: the location at ~325 m x-axis is moving with ~100 m downslope according to the results of the DIC here. Just ~25 m along the profile line in both directions, movement of less than 10 m has been tracked. How should this be possible, if the tracking did not simply produce random correlations with erroneous locations? What I am trying to say here is that the presentation of the results in the manuscript allows the reader to take this data for granted. Yet the authors show in their rebuttal to a number of my comments that they themsevles do not view these data as reliable. Please indicate clearly in the figures and the text that these data are erroneously matched pixels, or provide a different plausible explanation for the pattern. Furthermore, in my view the term decorrelated is misleading in the following way: technically, the patches that the DIC matches are correlated to another patch (see your reply to my comment L288/289), else no vector should be produced by the matching procedure. The pattern in these "decorrelated" areas of the landslide indicates that the matching did not find the corresponding patches and therefore does not produce a smooth displacement pattern, but random noise. In other words, despite the correlation the DIC finds, it correlates the wrong locations (images patches), i.e. produces wrong displacement vectors.
Thank you for your detailed feedback. We have revised the manuscript and defined *decorrelation* based on the current literature and our understanding. Accordingly, we have further indicated the ambiguous signals and explained their source (caption Fig. 5, caption Fig. 6, section 6.1.: L387f., L423ff., L430ff and section 6.2.: 470ff..). Please see also our comments further below.

[L404] Here we follow Leprince et al. (2007) describing a correlation loss as '*decorrelation*' with signal–to–noise values of low/null (i.e. no convergence of the correlation algorithm) and/or large offsets, either unrealistic in nature or beyond the valid matching window distance. Decorrelation in our understanding exhibits a salt–and–pepper appearance in the DIC result with random displacement vectors, related to inconsistently tracked features. The software is not able to find the corresponding, correlated surface pattern, leading to a misfit (i.e. misrepresentation) and/or mismatch (i.e. blunders) of the matching windows and finally resulting in noise (Debella-Gilo, 2011; Guerriero et al., 2020). Nevertheless, this decorrelation signal is still a valuable observation that might be related to surface processes and not only to erroneous limitations of the DIC method. There are three main reasons that might cause these effects: (i) significant temporal change of the surface, i.e. revolving and/or rotational deformation, (ii) high displacements exceeding the matching window size being smaller than the offset, (iii) land cover changes such snow cover, vegetation cover and alluvial processes, among others, and (iv) changes related to illumination (e.g. shadow) or image errors (e.g. orthorectification, shifts in individual bands) (Leprince, 2008; Debella-Gilo, 2011; Lucieer et al., 2014; Stumpf et al., 2016).Leprince (2008) describes snow cover, vegetation cover and alluvial processes, among others, as potential explanations for these decorrelations. In our study, the decorrelated areassalt–and–pepper areas include to a large degree the landslide head (a), the drainage channel (h) (Fig. 5a, b), a larger patch south of the active area boundary (g) (Fig. 5a), and some smaller ones in little depressions (g) (Fig. 5a) and (j) (Fig. 5a, b). Most The patches (*j*) and east of (*j*) are identified as snow fields in the orthophotos and the noise results from decorrelation.

Comment: 4
These are the points a) to d) I mentioned in my review statement.
Thank you, you are absolutely right. We removed the results and discussion of the 3 m DIC UAS analysis from the manuscript. Please see our detailed answer to comment 2, page 12 further below.

Comment: 5

Thank you very much for following a number of suggestions that sadly did not really improve the results of the image correlation.
Thank you for your suggestion to include pre-analysis calculations in the online supplementary material which helped to shed light on the complexity of the Sattelkar.

180   Comment: 1
Please, see my above comment on the idea of decorrelation.
Thank you, we have incorporated your comments #2 and #3, page 2.

Comment: 2
This is a crucial point. While you acknowledge here that in the area of decorrelated values, matching did in essence not work, this is not stated as explicitly in the manuscript. In my view, it would be necessary to clearly highlight the decorrelated areas as such in the Fig. 5 and explicitly mention the division in reliably tracked regions and those regions that are unreliable (i.e. random) and cannot be interpreted. In my impression this is not done in the present revised manuscript.

190   Thank you for pointing this out; now we understand your intentions in the first review round. In our revised manuscript we have clearly stated how these decorrelated areas should be interpreted. Please see also our previous comment (#2, page 2) with indicated manuscript changes. In addition, we have included a definition of "decorrelation" based on the current literature and our understanding.

[L422ff] However as field observations provide evidence that the rock masses are deforming, and the surface is altering due to the high mobility and rotational behaviour of some boulder blocks. This, which leads to changed pixel values and spectral characteristics of the block surface and the surrounding area, which can also result in poor correlations, and even random errors and mismatches (Debella-Gilo and Kääb, 2011). This finding is similar to observations in a rock glacier study by Debella-Gilo and Kääb (2011).

200

Comment: 3
This is very true and from a image processing point of view, it is the same reason why shadow and snow cover effects induce erroneous correlations. The changed pixel values inhibit rigorous matching. Please include more than just the speed of the landslide in your discussion of errors in matching between images - I also state that in another comment further.
Thank you for identifying this in our first letter of response and the first revised manuscript. We have modified the text accordingly. Please also see our answers below.

[L403ff] There are three main reasons that might cause these effects: (i) significant temporal change of the surface, i.e.
210   revolving and/or rotational deformation, (ii) high displacements exceeding the matching window size being smaller than the offset, (iii) land cover changes such snow cover, vegetation cover and alluvial processes, among others, and (iv) changes related to illumination (e.g. shadow) or image errors (e.g. orthorectification, shifts in individual bands) (Leprince, 2008; Debella-Gilo, 2011; Lucieer et al., 2014; Stumpf et al., 2016). Leprince (2008) describes snow cover, vegetation cover and alluvial processes, among others, as potential explanations for these decorrelations.

Comment: 4
I am very surprised that the authors try to find arguments for the validity of the data shown in Fig. 5 e and f. The results from the downsampled UAS give completely different displacements as the high-resolution UAS data, yet the data and time span are the same. How can the authors interpret these results as "less

220 trustworthy"? In interval II for example, the landslide either moves ~2 m (Fig. 5 b) or ~6 to >18 m (Fig. 5 f), but not both at the same time. These values are not similar in any kind of way, i.e. one of these results is simply wrong. I have been outlining this in detail above (highlighted lines of the general comment B raised in the frist round of reviews) and want to add an additional aspect to point the authors to the contrasting results these data suggest, if taken for granted: If you calculate a velocity from the displacement for e) and f), this translates to an average velocity of 17.67 m/yr for interval I as opposed to an average velocity of 156.5 m/yr for interval II, i.e. an acceleration of factor 8.9. The data for a) and b) however yield velocities of 9.71 m/yr and 17.39 m/yr, respectively, translating to a much more plausible acceleration of factor 1.79.

We thank you for emphasising this point. We discussed this again in detail and follow your arguments.
230 Therefore we have removed these results and revised the manuscript accordingly.

Comment: 1
Sorry for the typo...

Comment: 2
Exactly. That is why I commented that in theory, planet offers a big advantage here with daily data availability. But very practically, as you show with Tab. 2, only ~10% of the planet scenes were usable.
240 Effectively you end up with rather similar revisit times between planet and sentinel 1, if you want to use these examples here. But don't get me wrong, optical imagery has a lot of other advantages...

Thank you for this suggestion. We have modified the text accordingly.

[L510-514] Regardless of any meteorological constraints, the promised daily availability by PlanetScope is unrealistic, due to data gaps and provider issues; our study showed that for the Sattelkar from April to October 2019 only 11 % of the captured images during this time were usable. Hence, PlanetScope data ends up in a similar temporal availability to Sentinel–1 with a 6-day revisit time. In time–critical early warning scenarios, when time is running out, all available even partly usable images will be utilised and fieldwork may be conducted, even if the prevailing conditions are suboptimal but will increase data availability.

250

Comment: 1
Thank you for clarifying this. Please, include this in the manuscript and replace the ambiguous term "spectral colour problems" with the much more precise spatial offset or "shifts in the individual bands".

Thank you for this suggestion. We have modified the text accordingly.

[L267-269] Thereafter, a second selection (visually with the 'Map Swipe Tool' plugin) from the downloaded images was filtered for errors of location, inter-tile shift and shifts in the individual bands spectral colour problems which were
260 previously not clearly discernible in the online data hub.

Comment: 1
Thank you for the detailed answer. My comment was intended to motivate the authors not only to provide me with these details, but to include them in the revised version of the manuscript. Please include this explanation, or a short version of it, in the text. Regarding the residual mismatches stated here and shown

in Fig. 14 of the OSM: this already gives a measure of the amount of significant displacement, i.e. beyond a level of detection, that can be detected with DIC between these images. If your residual mismatch after coregistration (as shown in Fig. 14 OSM) averages to 0.6 - 0.8 pixel, everything below that cannot be treated as significant motion. But still, this would be much lower than your arbitrary 4 m and would be based on a preproducible quantification approach.

Thank you for this good suggestion. We have modified the text and added the information in the methods section of the manuscript.

[L462-464] This is clearer for the time interval I (376/370 d) (Fig. 5a vs. c) as for the longer temporal baseline the total displacement accumulation is higher, thus better captured by COSI–Corr for PlanetScope 3 m resolution. Due to the shorter interval II (42 d) (Fig. 5b vs. d) with less accumulated total displacement, the rear of the landslide is not represented; no signal is shown as the total displacement for PlanetScope was restricted to values above 4 m. Values below 4 m had to be discarded for PlanetScope DIC results as they were lost in noise, i.e. for the entire DIC results there is total displacement between 0 m and 4 m (cf. the online supplementary material (OSM) Fig. 13).

Comment: 2
Though in the current state the details of Fig. 3 b) and h) are difficult to identify, it is clear that the regions with smooth displacement value distributions show consistent bearings, while the regions with decorrelated displacements also show random bearing. In my experience, this is a clear sign of errors during the matching procedure, i.e. the correlation found highest agreement with the original feature. I would recomment to include the bearing information more prominantly in the manuscript itself, as this is a very important information on the data quality.

Thank you for this good suggestion. We have modified Fig. 5 by including the bearing information in the manuscript. The interpretation to this is discussed method text and caption. In addition we have added another map in the OSM showing the displacement vectors only (see. Fig. 17).

Comment: 1
I am unsure how the active area extent is related to the definition of the threshold value here?

Thank you for your comment. We apologise for our vague explanation. Besides the value distribution for both, the total displacement results and the SNR including their visual comparison, we used the demarcated active area as visual guidance to distinguish between values within the active area and aberrant displacements outside of it. Please see our comment below.

Comment: 2
It would be appropriate to include this description into the manuscript. As I pointed out in my comments in the first round of reviews, the error assessment is very important and just setting an arbitrary threshold of 4 m without elaborating the calculation of this values is insufficient.

Thank you, this is a good suggestion and we included it in our revised manuscript.

[L307ff]
Displacement below a 4 m threshold was discarded from the PlanetScope datasets due to aberrant values (noise, outliers). The threshold definition was defined on (i) the value distribution in both the total displacement and the corresponding SNR result, and (ii) a visual comparison of the maps for the total displacement and the SNR. This definition allowed us to identify outliers and unlikely displacement. Apart from this threshold; no other filters were employed, and we  kept the output raw (see for raw DIC on PlanetScope OSM Fig. 13).

Comment: 3
I strongly doubt that you can make this statement. The image processing algorithm matches pixel value distributions from patches in consecutive images. When the pixel value distribution changes between
320 images, the algorithm does not find the "true" corresponding area, but matches to the most similar patch it finds. Whether the pixel value distribution changes in reaction to snow cover, shadows, or vegetation, or if large displacements induce the spectral differences is not discernable for the DIC software.
We completely agree that the software cannot differentiate between the different reasons for the resulting displacement; hence the software returns matches for pixel changes of rediscovered reference patches independent of the reason for these changes. And yes, you are right we used the same input data here (UAS orthophotos) and compared two different orthophoto derivations to check for result behaviour and consistency.
What we intended to say is that the user may be able to determine if the orthophoto shows differences in illumination, vegetation or snow cover, or if volumetric calculations show significant surface elevation
330 changes (as visualised in the OSM Fig. 2 and manuscript Fig. 6).
We have revised our manuscript and described this in more detail.

[L469-473] Measured ground motion of block tracking and PlanetScope results indicate and support existing high ground motions. In addition there are morphologically significant volumetric turnovers with areas of large gains and losses between ± 5 m (see OSM Fig. 11). These observations might  explain the observed decorrelation at the finer resolution of 0.16 m for the landslide head: the matching window is smaller than the offset and texture surface changes are too complex to be re–detected, i.e. matched, and thus correlated, leading to decorrelation and noise. Homogeneous correlated patches are in the front of the landslide body for the shorter time interval; there may have been some displacement just below the detection threshold for this high ground motion or
340 some boulders and their surroundings might have been matched, or both (Fig. 6a (a)). . In this case for the complex ground motion with high spatial resolution data , the previous assumption  based on a shorter time interval likely leads to improved detection of inherent process behaviour (see Sect. 6.1.).

and

[L430ff] In sum, though the results contain heterogeneous, noisy, decorrelated areas, the combination with homogeneous displacement areas still offers valuable insights into this and other internal landslide structures and complex behaviours.

Comment: 4
While I do agree that visually tracking boulders in orthoimages gives reliable results that underpin the
350 displacement that was obtained with digital image correlation in the same images, the term verification implies that these displacements come from independent data.
Yes, you are right. To call it "verification" two independent data types would be necessary. Thank you, we understand what you meant and therefore changed the term accordingly to a better suited description.
Please see here also our answer to the previous comment (no. 3, p. 9).
Visual tracking of 36 single blocks, identifiable in the UAS orthophoto series allowed deriving direction and amount of movement; this supported the  confirmation process  for (i) the total displacement and (ii) the results of automated and manual tracking. In the next section we present this approach only for time interval II.

Comment: 1
This is a good idea. Sadly, but maybe owed to the reduced quality of the review files, I find it very hard to identify vectors shown in Fig. 3 b) and h). Also for the signal to noise ratio figures, shown in Fig. 3 d) and j), the transparent grey colours on top of the greyscale hillshade image are difficult to identify.
We are sorry that the results are difficult to identify and hope that the quality of the uploaded material is now higher.
In addition, we have added displacement vectors to Fig. 5 for the UAS DIC result interval II (13.07.2018–24.07.2019).

370

Comment: 1
In my view, it might be advisable to plot the manually tracked boulder displacements against the mean displacement obtained by your DIC for the surrounding pixels (and not just one pixel). As image correlation for tracking is based on matching of value distributions, i.e. matching patches of pixels, adjacent pixels should show similar displacement magnitude and bearing. With this, you could strengthen the argument that the boulder tracking actually backs the DIC tracking - maybe not for all regions, but for the majority.

380 Thank you for this good suggestion. We have calculated the mean total displacement from COSI-Corr of 0.1 m buffers around the manually tracked boulders and plotted them against the distance the boulders travelled (please see the OSM Fig. 15 and Fig. 16). The residual standard error is 0.7137 (on 31 degrees of freedom) with a multiple R-squared of 0.7425.

Comment: 1
I don't think that the comment on L457/462, where you limit your answer to areas outside the landslide area is related to the issue raised here.
390 Thank you for this comment.

Comment: 2
It is my impression that the authors have understood my above comment differently than it was intended. Figs. 5 a) and e) as well as b) and f) show the results of the first and second interval, respectively. While a) and e) result from DIC of 0.16 m resolution orthoimages, b) and f) result from the same images, but resampled to 3m. Yet the pattern and magnitude of e) and f) do in no way match the data in a) and b), while it is the same data over the same time interval. My point is that this does not make sense from a technical point of view when automatically tracking features in the same images, with only a different resolution. I would think that the authors need to decide, which of the two solutions is closer to the truth,
400 but in my view it has to be made absolutely clear in the manuscript that these data do not match. When pointing out that I do not see a plausible explanation for the data presented in e) and f) from a geomorphic point of view, I was hoping for a critical evaluation of this data.
You are right. We discussed this matter and agree with the points you mentioned. Therefore we deleted the row in Table 6, modified Fig. 5 including its caption and revised the methods, result and discussion section accordingly: L313-314, 317-318, 322-323, L352-359, L457-460 and L467-470 were deleted, L460-462 were revised and extended.

Comment: 3
Again, I fear this misses the point. In the text you state that false-positive displacements were observed
410 and I was just interested in the analysis conducted to identify these. Please elaborate.

Thank you for emphasising this. We have investigated the orthophotos and satellite images in detail. Please see our analyses in the OSM Fig. 2, 11 and 12. We have modified the text accordingly.

[L407 ff] Most The patches (j) and east of (j) are identified as snow fields in the orthophotos and the noise results from decorrelation. In Fig. 5a, the large southern patch (g) shows clear displacement values for the rear part and decorrelation for the front region resulting from significant morphological changes within an the image pair of interval I (see OSM Fig. 12). This is due to a gain between 1 and 2 m for an area of about 250 m².

420   Further, the second comment refers to the general statement you are trying to make here from your analysis. As mentioned before, there are a lot of approaches to enhance tracking results and I would urge the authors to be very cautious in attesting planet imagery a limited use here, given the challenging results obtained.
Thank you for clarifying this.

Comment: 1
I have to express a certain degree of frustration with this review, as the author's tendency to rebuttal seems
430   higher than their inclination to improve their manuscript: in Line 1 (the tiltle of your manuscript) the word LANDSLIDE is misspelled. And it is still misspelled in the title of the revised manuscript and on the NHESS webpage.
We are very sorry for this spelling mistake. None of us noticed it through revising the manuscript multiple times prior submitting it to NHESS. We immediately notified NHESS about this typo and asked for it to be corrected on the website and corrected it in our manuscript, too.
We apologise as we misinterpreted what you meant by 'landslide' in your first review, assuming you wanted an explanation and better description of our understanding of the Sattelkar landslide.